# Natural rhythms of periodic temporal attention

Arnaud Zalta [1,2], Spase Petkoski [1] & Benjamin Morillon [1✉]

That attention is a fundamentally rhythmic process has recently received abundant empirical evidence. The essence of temporal attention, however, is to flexibly focus in time. Whether this function is constrained by an underlying rhythmic neural mechanism is unknown. In six interrelated experiments, we behaviourally quantify the sampling capacities of periodic temporal attention during auditory or visual perception. We reveal the presence of limited attentional capacities, with an optimal sampling rate of ~1.4 Hz in audition and ~0.7 Hz in vision. Investigating the motor contribution to temporal attention, we show that it scales with motor rhythmic precision, maximal at ~1.7 Hz. Critically, motor modulation is beneficial to auditory but detrimental to visual temporal attention. These results are captured by a computational model of coupled oscillators, that reveals the underlying structural constraints governing the temporal alignment between motor and attention fluctuations.

[1] Inserm, INS, Inst Neurosci Syst, Aix Marseille University, 13005 Marseille, France. [2] APHM, INSERM, Inst Neurosci Syst, Service de Pharmacologie Clinique et Pharmacovigilance, Aix Marseille University, 13005 Marseille, France. ✉email: bnmorillon@gmail.com

Adapting our behaviour according to external stimuli requires extraction of relevant sensory information over time[1]. This ability relies on the capacity to flexibly adapt and adjust our temporal attention to the natural dynamics of the environment. For instance, sailing on tumultuous seas, following the flow of an animated speaker or listening to drums in an ebullient jazz band requires specific tuning of our temporal attention capacities. But in some cases, this ability fails. A stream of events appearing too fast or in a temporally disorganised way are situations typically difficult to attend to. Multiple types of temporal structures are capable of guiding temporal attention[1], such as isochronous[2] or heterochronous streams of events[3], symbolic cues[4], or hazard functions[5]. Paradigms that involve isochronous perceptual streams in the auditory and/or visual modality are consistently designed with rhythms in the 1–2 Hz frequency range[2,6–27], which, incidentally, corresponds to the natural musical beat frequency—or tempo[28]. Strikingly, the propensity to flexibly focus in time and its limits—in other words, the sampling capacities of temporal attention—have never been investigated.

Contrary to the continuous flow of perceptual events, actions are coordinated and rhythmic. For instance, walking is intrinsically rhythmic and operates at ~2 Hz[29]. Spontaneous motor rhythmic behaviours such as finger tapping also operate at a preferred tempo of ~1.5–2 Hz and motor tapping has an optimal temporal precision within the range of 0.8–2.5 Hz[30–32]. Moreover, delta (0.5–4 Hz) neural oscillations shape the dynamics of motor behaviour and motor neural processes[33]. For instance, during production of complex motor behaviours such as speech, the coordination of articulatory movements is encoded in kinematic trajectories characterised by damped oscillatory dynamics[34]. Even during non-periodic motor behaviours, such as reaching, motor trajectories are encoded in neural dynamical patterns that oscillate around 1–2 Hz[35–37]. And crucially, the motor cortex exhibits resting-state dynamics at the delta rate[36,38]. Thus, delta oscillations are an intrinsic rhythm of the motor system visible in the dynamics of most basic motor acts.

In line with the active sensing framework, perception involves motor sampling routines like sniffing and whisking in rodents or visual search in primates[39–41]. Attention is an essential component of this process, its influence helping to impose the motor sampling pattern on the relevant sensory stream[21,40]. Accordingly, previous studies showed that overtly moving during an auditory attention task improves perceptual performance[21,22]. Importantly, these experiments were also performed at 1.5 Hz, which both corresponds to the rhythm classically used to investigate periodic temporal attention and to the natural rate of rhythmic movements. In virtue of the fundamental relationship between motor and active/attentive sensory processes, one could hypothesise that the sampling capacities of periodic temporal attention derive from those of the motor system and are thus limited around 1.5 Hz. Alternatively, one could hypothesise that temporal attention is not rate-restricted but that it is the motor benefit to temporal attention that is restricted around this rate. Finally, the rhythmic sampling rate of visual sustained attention was recently shown to be restricted around 4–8 Hz[42,43] which suggests that sensory-specific temporal constraints could also shape the sampling capacities of temporal attention.

To investigate these issues, we developed a paradigm to behaviourally quantify the sampling capacities of periodic temporal attention during auditory and visual perception. The quality of temporal attention was estimated for different rhythms, ranging from ~0.5 to ~4 Hz. In each modality, we first investigated temporal attention during passive perception—i.e., without overt motor involvement—and then quantified in another set of experiments the motor contribution to temporal attention.

Through six interrelated behavioural experiments, we reveal the existence of a limited sampling capacity of temporal attention, which is moreover sensory-specific. Besides, we highlight that the motor contribution to temporal attention is also sensory-specific and derives from the compatibility of temporal dynamics underlying motor and sensory-specific attentional processes. Finally, in line with previous models of beat perception and temporal attention processes[23,24,44–46], we show that our results are reproduced by a simple model involving three coupled oscillators. While the optimal sampling rate of temporal attention is directly reflected in the natural frequency of the attentional oscillator, the quality of the motor modulation crucially depends on the time-delay in the coupling between the stimulus and the motor oscillator.

## Results

**Rhythmic sampling capacities of temporal attention.** The tasks of this study are all based on the same paradigm. Sequences of stimuli were presented on each trial, from 2 to ~20 s. Three reference stimuli defining the tempo (or beat frequency) of the isochronous event sequence preceded a mixture of on-beat and off-beat stimuli. Participants performed a beat discrimination task at the end of each trial, by deciding whether the last stimulus of the sequence, a deviant, was on or off beat (Fig. 1a). While on-beat stimuli were reinforcing the beat, crucially, off-beat stimuli had a distracting influence. This interleaved delivery of sensory events forced participants to track the beat throughout the entire duration of the sequence while minimising the interference of aperiodic events. This protocol thus ensured that their attentional focus was temporally modulated over an extended time period. The density of distractors (i.e., number of distractors per beat) was adjusted for each participant prior to the experiment to reach threshold performance for a 2 Hz beat-frequency (see Methods section). The beat frequency varied from ~0.5 Hz to 3.8 Hz across conditions, to span most of the range of discernible tempi[47,48].

**An optimal rate for auditory periodic temporal attention.** In a first passive auditory experiment (exp. 1) we used pure tone stimuli. Eight conditions were investigated with isochronous tempi of 0.6, 0.7, 1, 1.3, 1.7, 2.2, 2.9 and 3.8 Hz (Fig. 1c; see Methods section). The average difficulty level (density of distractors) was around 1 ($M = 1.01$; $SD = 0.70$; Fig. 1b). The comparison of conditions revealed significant fluctuations in performance (% correct responses) across tempi (repeated-measures ANOVA: $F(7,203) = 15.3$, $p < 0.001$; Fig. 1c). The profile of performance across tempi had moreover an inverse U-shape profile, which could be properly approximated with a third-degree polynomial function ($R^2(8) = 0.86$, $p = 0.002$; see Methods section), whose local maximum estimates the tempo at which performance is optimal. Estimates of the optimal tempo measured with individual fits revealed that auditory temporal attention has an optimal rhythmic sampling frequency of ~1.3 Hz ($M = 1.34$ Hz; $SD = 0.80$ Hz; Fig. 1d).

In this experiment, tones lasted 10% of the beat period (see Methods section). Our results could thus be due to the existence of either an optimal beat frequency or an optimal stimulus duration, during auditory temporal attention. In a subsequent control experiment, we thus orthogonalized tempo and stimulus duration, by fixing across conditions tones length to 22.5 ms (Supplementary Fig. 1). We replicated all findings of experiment 1 (repeated-measures ANOVA: $F(7,98) = 8.9$, $p < 0.001$; difficulty level: $M = 0.95$; $SD = 0.63$; 3rd order fit quality: $R^2(8) = 0.86$, $p = 0.002$; individual estimates of optimal tempo: $M = 1.32$ Hz; $SD = 0.48$ Hz). This indicates that fluctuations of performance across

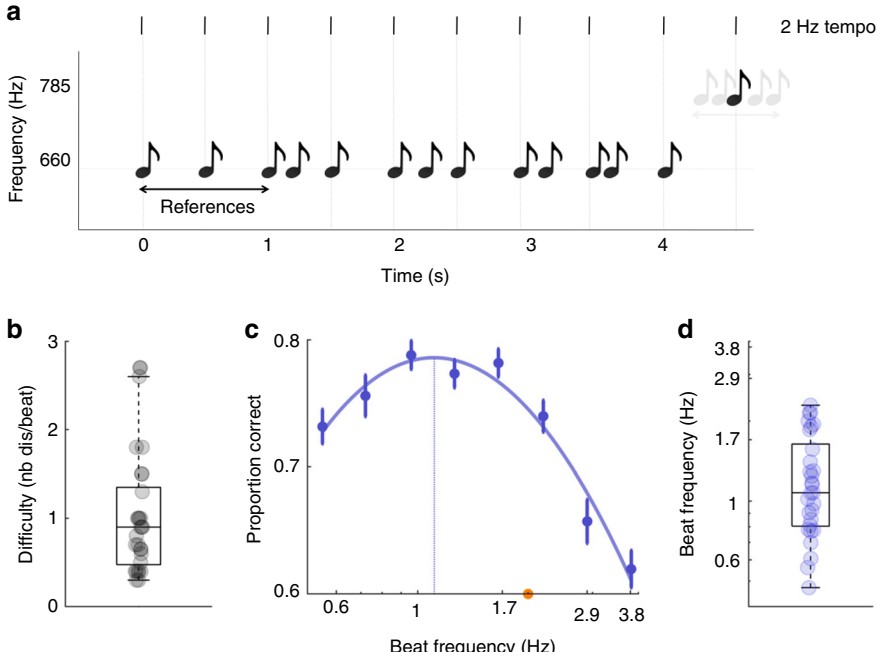

**Fig. 1 Passive auditory experiment: experimental design and results.** Experiment 1. **a** Sequences of pure tones were presented binaurally on each trial. Three reference tones defining the beat frequency (or tempo; vertical lines) of the sequence preceded a mixture of on-beat and off-beat tones. Participants performed a beat discrimination task at the end of each trial, by deciding whether the last tone of the sequence, a spectral deviant (785 Hz vs. 660 Hz), was on or off beat. The beat frequency varied across conditions (from 0.6 Hz to 3.8 Hz). **b** Individual difficulty level to reach threshold performance for a 2 Hz tempo. Difficulty was modulated by adjusting the density of distractors in the sequence (number of distractors per beat). **c** Average performance per condition (beat frequency). Data were approximated with a polynomial function (plain line), and an optimal tempo (leading to a maximal performance) could be estimated (vertical line). The orange dot indicates the tempo (2 Hz) that served to estimate the individual difficulty level. **d** Individual estimates of the optimal tempo. One outlier participant (4.5 Hz) is not visible. Error bars indicate s.e.m. ($n = 30$). Boxplots represent median and 1.5 times the interquartile range.

conditions are due to the existence of an optimal tempo at which auditory temporal attention operates.

We also investigated whether this result could be explained by the position of the deviant relative to the beat. We conducted a GLMM analysis, in which the nature of the deviant (on-beat or off-beat) and its distance relative to the beat (in ms, in relative value) were entered as predictors, in addition to the beat frequency. These three variables were not correlated (all $R^2 < 0.13$), and each of them significantly contributed to the model (nature of deviant: $X^2 = 18.2$; distance: $X^2 = 13.2$; beat frequency: $X^2 = -11.2$; all $p < 0.001$). This indicates that while the position of the deviant impacted performance accuracy, it could not explain the fluctuations of performance observed across tempi.

**Motor contribution to auditory temporal attention.** In a second experiment (exp. 2), we investigated whether motor activity helps to synchronise temporal fluctuations of attention with the timing of events in a task-relevant stream. Participants carried out two sessions. A 'passive' one where they performed the task while staying completely still for the duration of the trial (as in exp. 1), and a 'tracking' one, where they hit the beat in phase with their index finger on a noiseless pad. Therefore, the absence or presence of overt movement was the single difference between the two sessions. While it is not possible to control for the covert involvement of motor and/or premotor structures during temporal attention tasks, comparing passive and tracking sessions allowed us to quantify the influence of overt (relative to covert) motor activity on the precision of temporal attention.

We observed significant fluctuations in performance across tempi (repeated-measures ANOVA, condition: $F(7,133) = 15.9$, $p < 0.001$; Fig. 2a). The comparison of passive and tracking

sessions revealed a significant difference in categorisation performance (session: $F(1,19) = 7.5$, $p = 0.013$), which was moreover beat-selective (interaction: $F(7,133) = 2.8$, $p = 0.023$). Post-hoc t-tests indicated that overt motor tracking significantly increased performance only when participants performed the task between 1.3 and 2.2 Hz (paired t-tests, 1.3 Hz: $p = 0.013$, $t(19) = 2.75$; 1.7 Hz: $p = 0.009$, $t(19) = 2.93$; 2.2 Hz: $p = 0.002$, $t(19) = 3.59$; all other beats: $t(19) < 1.95$, $p > 0.05$). The inverse U-shape profile of performance could be properly approximated with a 3rd order fit for both sessions (passive: $R^2(8) = 0.73$, $p = 0.011$; tracking: $R^2(8) = 0.91$, $p = 0.001$). The optimal beat frequency estimated with individual fits was around 1.5 Hz in both sessions (passive: $M = 1.47$ Hz; $SD = 0.59$ Hz; tracking: $M = 1.47$ Hz; $SD = 0.45$ Hz; passive vs. tracking: paired t-test: $t(19) = 0.02$, $p = 0.99$; Fig. 2d). To evaluate the likelihood that this absence of difference across sessions corresponds to a genuine absence of difference, we computed the corresponding Bayes factor (see Methods section). We obtained a Bayes factor of 0.22 for this null effect, providing significant evidence for the "null" hypothesis (no difference of optimal beat frequency between sessions). Of note, the optimal beat frequency was also not significantly different across the two auditory experiments (exp. 1. vs. passive exp. 2: unpaired welsh t-tests: $t(48) = 0.66$, $p = 0.51$, Bayes factor = 0.34). These results confirm previous findings showing that overt motor activity optimises auditory temporal attention[21,22] and further reveal that this benefit is rate-restricted and maximal around 1.5 Hz.

**Similar optimal rates for motor tapping and auditory temporal attention.** To further investigate the nature of the interaction between motor activity and auditory attention, in a third study we

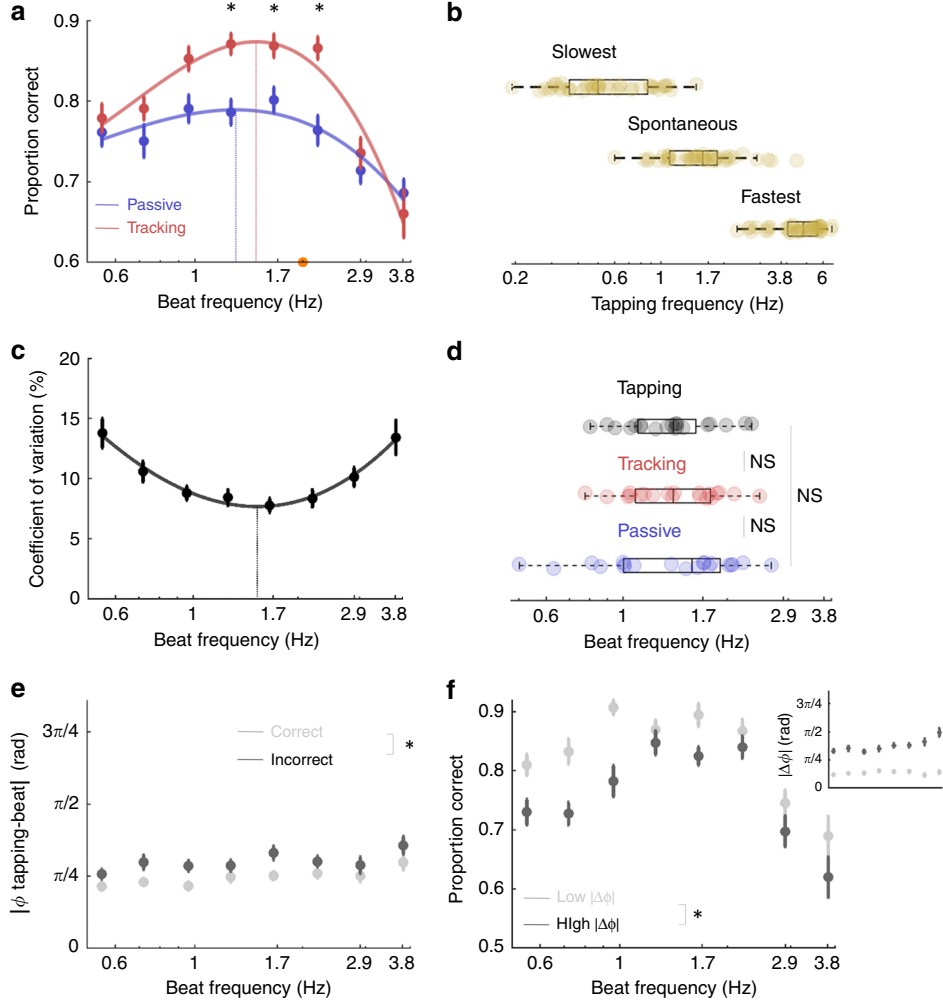

**Fig. 2 Motor contribution to auditory temporal attention and free tapping. a** Experiment 2. Average performance per condition (beat frequency) in the passive (blue) and tracking (red) sessions. In the passive session, participants performed the task without moving before the end of the sequence. In the tracking session, participants performed the task while expressing the tempo by moving their index finger. Same conventions as in Fig. 1c. **b** Free tapping experiment (exp. 3). Individual average frequency of free tapping in three conditions where participants had to rhythmically tap at their slowest, spontaneous and fastest rate. **c–f** Experiment 2: **c** Coefficient of variation (CV; i.e., relative standard deviation) of guided tapping across conditions in the tracking session. Same conventions as in Fig. 1c. **d** Individual estimates of the optimal tempo of auditory temporal attention in the passive (blue) and tracking (red) sessions (from A) and of guided motor tapping in the tracking session (black; from c). **e** Sensorimotor simultaneity (Φ; in radian), i.e., temporal distance between motor acts and the beat (in absolute value, normalised to the beat period) across conditions in the tracking session, for correct (light grey) and incorrect (dark grey) trials. **f** Average performance per condition in the tracking session for trials with low (light grey) and high (dark grey) sensorimotor simultaneity indexes. Trials were sorted according to a median-split procedure. The inset plot indicates the associated sensorimotor simultaneity indexes. Error bars indicate s.e.m. ($n = 20$; (**a**) paired $t$-tests, (**e–f**) repeated-measures ANOVA; $*p < 0.05$; NS non-significant). Boxplots represent median and 1.5 times the interquartile range.

asked participants to perform a standard free tapping experiment (BAASTA[49]; see Methods section). In the absence of any sensory cue, they naturally tapped on average at ~1.7 Hz (spontaneous tapping frequency: $M = 1.67$ Hz; $SD = 0.74$ Hz; Fig. 2b), which confirms previous studies[29,32,50]. We also instructed participants to tap as slow and as fast as possible and found that the range of producible taps (slow: $M = 0.60$ Hz; $SD = 0.30$ Hz; fast $M = 4.69$ Hz; $SD = 1.13$ Hz) was similar to the range of discernible beats, i.e., ~0.5–4 Hz[47,48].

Furthermore, we analysed the guided motor tapping precision of participants across conditions during the tracking session of experiment 2. Participants tended to tap too fast during perception of the slowest tempi, too slowly for the fastest ones and tapped at the appropriate pace during presentation of a ~1.7 Hz beat frequency (Supplementary Fig. 2a). The coefficient

of variation (CV; i.e., relative standard deviation) of guided tapping across conditions confirmed that the quality of tapping differed across conditions (repeated-measures ANOVA: $F(7,133) = 6.84$, $p < 0.001$; Fig. 2c). It had a U-shape profile across conditions which could be properly approximated with a 3rd order fit ($R^2(8) = 0.95$, $p < 0.001$). Strikingly, individual estimates of the tempo associated to an optimal guided tapping rhythmicity ($M = 1.42$ Hz; $SD = 0.44$ Hz; Fig. 2d) were overall similar to the optimal tempi of auditory temporal attention in both passive and tracking sessions (paired $t$-tests, guided tapping vs. passive: $t(19) = 0.32$, $p = 0.75$, Bayes factor = 0.24; guided tapping vs. tracking: $t(19) = 0.44$, $p = 0.66$, Bayes factor = 0.25). Again, Bayes factor values provide significant evidence for the "null" hypothesis (no difference of optimal tempi). Thus, the optimal frequency of rhythmic movements in the absence or presence of synchronous

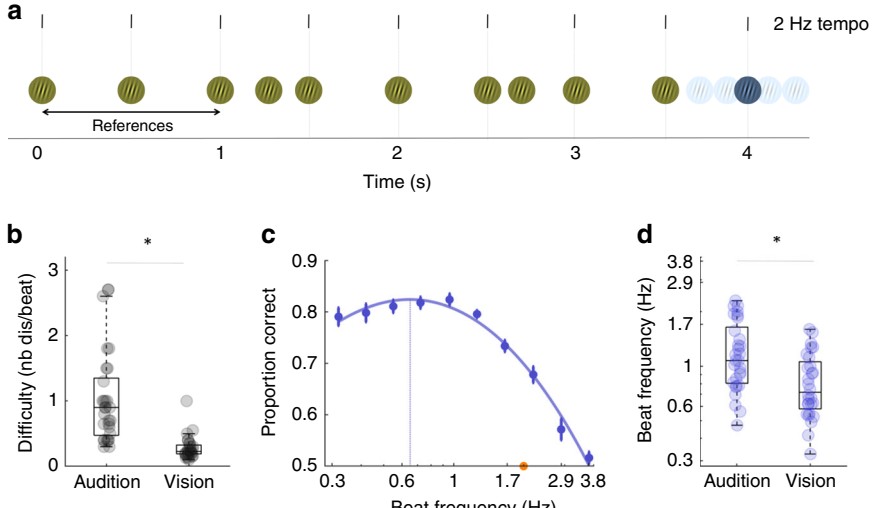

**Fig. 3 Passive visual experiment: experimental design and results.** Experiment 4. **a** In contrast to auditory experiment 1, sequences of visual gratings were presented on each trial. Participants performed a beat discrimination task by deciding whether the last grating of the sequence, a colour deviant (blue vs. yellow), was on or off beat. The beat frequency varied across conditions (from 0.3 Hz to 3.8 Hz) **b** Comparison of the individual difficulty levels obtained during the staircase procedures (with a 2 Hz tempo) of the auditory (exp. 1) and visual (exp. 4) experiments. **c** Average performance per condition. Same conventions as in Fig. 1c. **d** Comparison of individual estimates of the optimal tempo, between auditory (exp. 1) and visual (exp. 4) experiments. Error bars indicate s.e.m. ($n = 30$; unpaired $t$-tests; *$p < 0.05$). Boxplots represent median and 1.5 times the interquartile range.

periodic auditory cues is ~1.5 Hz, which is also similar to the optimal frequency of auditory temporal attention (in both the absence and presence of concomitant rhythmic movements).

**The quality of motor tracking positively impacts auditory performance.** To further explore these results, we investigated whether the quality of guided motor tapping influenced the quality of auditory temporal attention on a trial-by-trial basis. First, we compared the CV of guided tapping for correct and incorrect trials of the tracking session (Supplementary Fig. 2b). We observed an absence of difference between trials in which participants responded correctly or incorrectly (repeated-measures ANOVA, correct vs. incorrect: $F(1,19) = 0$, $p = 0.99$, Bayes factor $= .22$; condition $F(7,133) = 7.09$, $p < 0.001$; interaction: $F(7,133) = 1.14$, $p = 0.34$). We also compared the performance of participants in trials where the guided tapping CV was low or high, by using a median-split procedure (Supplementary Fig. 2c inset). Again, while the CV in these two groups of trials was highly different (repeated-measures ANOVA, low vs. high CV: $F(1,19) = 79.7$, $p < 0.001$; condition: $F(7,133) = 6.82$, $p < 0.001$; interaction: $F(7,133) = 4.41$, $p = 0.003$), we observed similar performance between these two groups of trials (repeated-measures ANOVA, low vs. high CV: $F(1,19) = 0.42$, $p = 0.52$, Bayes factor $= 0.28$; condition: $F(7,133) = 16.1$, $p < 0.001$; interaction: $F(7,133) = 0.47$, $p = 0.79$; Supplementary Fig. 2c). Overall, these results indicate that while the optimal rate of rhythmic movements and of auditory temporal attention is similar on average, there is no direct mechanistic relation between the quality of motor tracking and of auditory temporal attention.

Second, we investigated the temporal distance between motor acts and the beat, i.e., the degree of simultaneity of motor acts relative to on-beat tones (sensorimotor simultaneity, see Methods section). Participants tended to anticipate the beat in this modality, except when the tempo was too fast (≥2.9 Hz; Fig. S4a). We observed an overall better sensorimotor simultaneity in trials where participants' temporal attention was accurate than in incorrect trials (repeated-measures ANOVA, correct vs. incorrect: $F(1,19) = 23.4$, $p < 0.001$; condition $F(7,133) = 1.86$, $p = 0.15$; interaction: $F(7,133) = 0.79$, $p = 0.55$; Fig. 2e). We also split trials

in which the temporal distance between motor acts and on-beat tones was low or high (Fig. 2f inset; repeated-measures ANOVA, low vs. high: $F(1,19) = 209$, $p < 0.001$; condition: $F(7,133) = 2.3$ $p = 0.098$; interaction: $F(7,133) = 11.84$ $p < 0.001$). We observed a significant difference of performance between these two groups of trials (repeated-measures ANOVA, low vs. high: $F(1, 19) = 30.81$, $p < 0.001$, condition: $F(7,133) = 16.13$, $p < 0.001$; interaction: $F(7,133) = 1.62$, $p = 0.16$; Fig. 2f), indicating that the ability of participants to closely track the auditory tempo, vis. the quality of motor tracking, directly benefits performance accuracy.

**An optimal rate for visual periodic temporal attention.** In a first visual passive experiment (exp. 4), we used visual grating stimuli (Fig. 3a). Ten conditions were investigated with iso-chronous tempi of 0.3, 0.4, 0.6, 0.7, 1, 1.3, 1.7, 2.2, 2.9 and 3.8 Hz (Fig. 3c; see Methods section). Two participants did not complete the experiment and were excluded. The average difficulty level (density of distractors) was around 0.3 ($M = 0.28$; $SD = 0.18$), significantly lower than in the auditory tasks (comparison of exp. 1 and 4: unpaired welsh $t$ test, $t(56) = -5.52$, $p < 0.001$; Fig. 3b). In other words, fewer distractor had to be inserted per beat in the visual as compared to the auditory sequences to obtain similar levels of performance. The comparison of conditions revealed significant fluctuations in performance across tempi (repeated-measures ANOVA, condition: $F(9,243) = 53.6$, $p < 0.001$; Fig. 3c). They moreover had an inverse U-shape profile (3rd order fit: $R^2$ $(10) = 0.93$, $p < 0.001$). The estimated local maximum of the individual level performance revealed that visual temporal attention has an optimal rhythmic sampling frequency of ~0.8 Hz ($M = 0.83$ Hz; $SD = 0.34$ Hz; Fig. 3d). These results also reveal different preferred sampling rates of temporal attention among sensory modalities, with a significantly lower optimal beat frequency in the visual as compared to the auditory modality (comparison of individual estimates of the optimal tempo in exp. 1 and 4: unpaired welsh $t$ test: $t(56) = 3.18$, $p = 0.003$; Fig. 3d).

**Disruptive motor contribution to visual temporal attention.** In a second visual experiment (exp. 5), we investigated the motor influence on visual temporal attention across 8 conditions (Fig. 4;

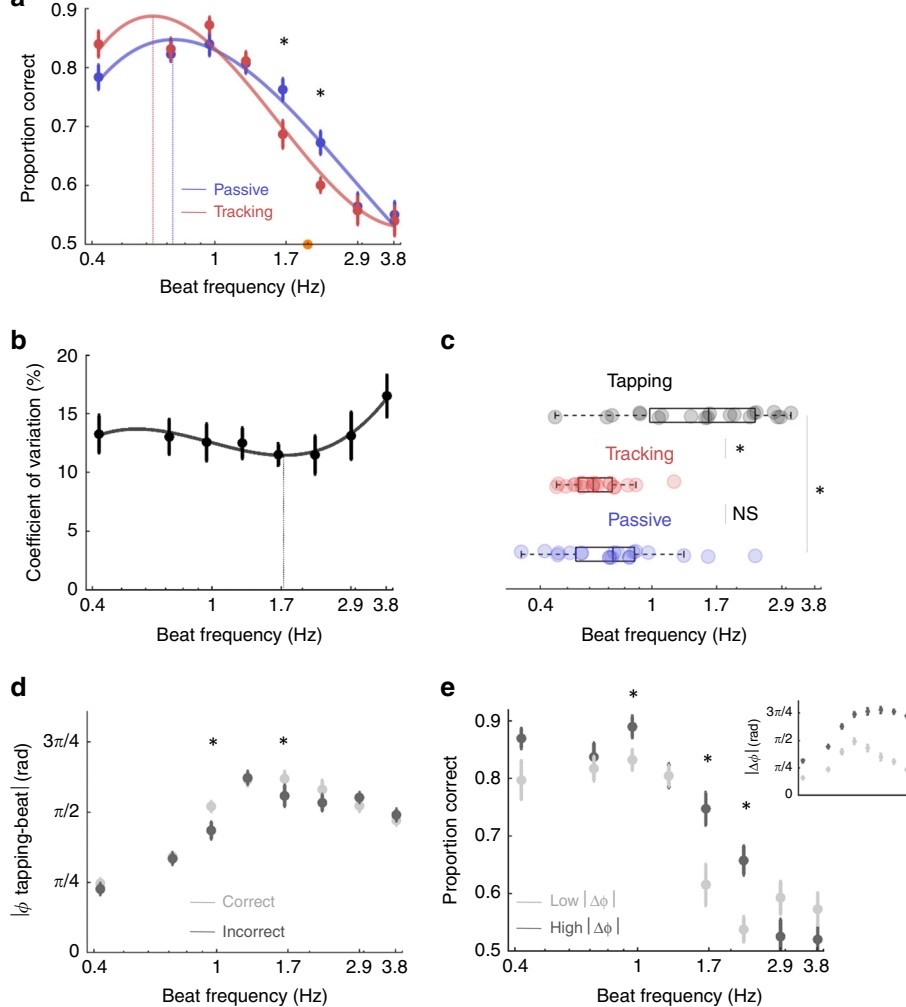

**Fig. 4 Motor contribution to visual temporal attention.** Experiment 5. **a** Average performance per condition in the passive (blue) and tracking (red) sessions. **b** Coefficient of variation (CV) of guided tapping across conditions in the tracking session. **c** Individual estimates of the optimal tempo of visual temporal attention in the passive (blue) and tracking (red) sessions (from **a**) and of motor tapping in the tracking session (black; from **b**). **d** Sensorimotor simultaneity (Φ; in radian) across conditions in the tracking session, for correct (light grey) and incorrect (dark grey) trials. **e** Average performance per condition in the tracking session for trials with low (light grey) and high (dark grey) sensorimotor simultaneity indexes. The inset plot indicates the associated sensorimotor simultaneity indexes. Same conventions as in Fig. 2. Error bars indicate s.e.m. ($n = 20$; paired $t$-tests; *$p < 0.05$; NS non-significant). Boxplots represent median and 1.5 times the interquartile range.

see Methods section). We first observed significant fluctuations in performance across tempi, which replicates the results of experiment 4 (repeated-measures ANOVA, condition: $F_{(7,133)} = 62.9$, $p < 0.001$; Fig. 4a). The comparison of passive and tracking sessions did not reveal a significant difference in overall categorisation performance (session: $F_{(1,19)} = 0.56$, $p = 0.46$) but a tempo-selective significant difference (interaction: $F_{(7,133)} = 2.8$, $p = 0.03$). In contrast to auditory perception, post-hoc t-tests indicated that overt motor tracking significantly decreased performance when participants performed the task between 1.66 and 2.2 Hz (paired $t$-tests: 1.66 Hz: $t_{(19)} = -2.23$, $p = 0.038$; 2.20 Hz: $t_{(19)} = -2.67$, $p = 0.015$; all other tempi: $t_{(19)} < 1.91$; $p > 0.05$). Like in the previous experiments, the inverse U-shape profile of performance could be properly approximated with a 3rd order fit for both sessions (passive: $R^2_{(8)} = 0.95$, $p < 0.001$; tracking: $R^2_{(8)} = 0.86$, $p = 0.002$). The optimal beat frequency estimated with individual fits was around 0.7 Hz in both sessions (passive: $M = 0.83$ Hz; $SD = 0.46$ Hz; tracking: $M = 0.65$ Hz; $SD = 0.17$ Hz; passive vs. tracking: paired $t$-tests: $t_{(19)} = -1.47$, $p = 0.16$; Bayes factor $= 0.66$; Fig. 4c). Of note, the optimal beat

frequency was also not significantly different across the two visual experiments (exp. 3. vs. passive exp. 4: unpaired welsh t-test: $t_{(46)} = -.07$, $p = 0.95$, Bayes factor $= 0.29$).

**Divergent optimal rates for motor tapping and visual temporal attention.** We analysed the guided tapping precision of participants across conditions during the tracking session of experiment 5, which indicated that participants tended to tap too fast in all conditions (Fig. S3a). Like in the previous auditory experiment (exp. 2), the CV of guided tapping had a U-shape profile (3rd order fit: $R^2_{(8)} = 0.78$, $p = 0.007$) and individual estimates of the tempo associated to an optimal tapping rhythmicity were not significantly different across modalities (vision: $M = 1.40$ Hz, $SD = 0.77$ Hz; audition: $M = 1.42$ Hz, $SD = 0.44$ Hz; unpaired welsh t tes: $t_{(38)} = 0.12$, $p = 0.91$; Bayes factor $= 0.31$). However, in the visual modality the CV of guided motor tapping was not significantly different across conditions (repeated-measures ANOVA, condition: $F_{(7,133)} = 0.88$ $p = 0.48$) and overall the tapping CV was significantly lower in the auditory than the visual

modality (comparison of CV averaged across comparable conditions, i.e., between 0.7 and 3.8 Hz; unpaired welsh t test: $t(38) = 2.18$, $p = 0.037$).

Counter to the auditory experiment, the optimal tempo for guided motor tapping was statistically different to the optimal tempo of visual temporal attention, in both passive and tracking sessions (paired t tests: guided tapping vs. passive: $t(19) = 2.93$, $p = 0.01$; guided tapping vs. tracking: $t(19) = 4.32$, $p < 0.001$; Fig. 4c). Thus, the optimal frequency of rhythmic movements in the presence of synchronous periodic visual stimuli reflects natural motor dynamics (~1.5 Hz) but differs from the optimal frequency of visual temporal attention which is ~0.7 Hz (in both presence and absence of concomitant rhythmic movements).

**The quality of motor tracking negatively impacts visual performance.** As in the auditory experiment, we compared the CV of guided tapping for correct and incorrect trials of the tracking session (Supplementary Fig. 3b) and observed an absence of difference between trials where participants responded correctly or incorrectly (repeated-measures ANOVA, correct vs. incorrect: $F(1,19) = 1.31$, $p = 0.27$; condition $F(7,133) = 0.85$, $p = 0.49$; interaction: $F(7,133) = 0.14$, $p = 0.89$; Bayes factor = 0.44). We also compared the performance of participants in trials where the tapping CV was low or high (Supplementary Fig. 3c inset; repeated-measures ANOVA, low vs. high CV: $F(1,19) = 48.5$, $p < 0.001$; condition: $F(7,133) = 0.88$, $p = 0.48$; interaction: $F(7,133) = 0.62$, $p = 0.61$), and observed similar performance between these two groups of trials (repeated-measures ANOVA, low vs. high CV: $F(1,19) = 0.06$, $p = 0.81$; condition: $F(7,133) = 41.8$, $p < 0.001$; interaction: $F(7,133) = 0.33$, $p = 0.87$; Bayes factor = 0.23; Supplementary Fig. 3c).

Investigation of the temporal distance between motor acts and the beat revealed that in the visual modality participants were not anticipating on-beat stimuli but tapped in reaction to it (Supplementary Fig. 4b). In other words, participants tended to tap both faster (Supplementary Fig. 3a) and later (Supplementary Fig. 4b) than the beat. In contrast to the auditory modality, correct trials were moreover associated with a lower degree of sensorimotor simultaneity than incorrect trials (Fig. 4d; repeated-measures ANOVA, correct vs. incorrect: $F(1,19) = 8.41$, $p = 0.009$; condition $F(7,133) = 31.5$, $p < 0.001$; interaction: $F(7,133) = 4.02$, $p = 0.004$). These effects were most pronounced at 1 and 1.7 Hz (post-hoc paired $t$-tests: 1 Hz: $t(19) = 3.25$, $p = 0.004$; 1.7 Hz: $t(19) = 2.4$, $p = 0.026$; all other tempi: $t(19) < 2.03$; $p > 0.056$). Splitting trials in which sensorimotor simultaneity was low or high (Fig. 4e inset; repeated-measures ANOVA, low vs. high: $F(1,19) = 796$, $p < 0.001$; condition: $F(7,133) = 33.1$ $p < 0.001$; interaction: $F(7,133) = 81$ $p < 0.001$) revealed a significant difference of performance between these two groups of trials (Fig. 4e; repeated-measures ANOVA, low vs. high: $F(1,19) = 6.2$, $p = 0.022$; condition: $F(7,133) = 41.76$, $p < 0.001$; interaction: $F(7,133) = 4.56$, $p = 0.002$). The ability of participants to closely track the beat was detrimental to performance accuracy, and this effect was most pronounced at 1, 1.7 and 2.2 Hz (post-hoc paired $t$-tests: 1 Hz: $t(19) = -2.93$, $p = 0.009$; 1.7 Hz: $t(19) = -2.59$, $p = 0.018$; 2.2 Hz: $t(19) = -2.76$, $p = 0.012$; all other tempi: $t(19) < 1.82$; $p > 0.084$). These results elucidate the observed disruptive motor contribution to visual temporal attention (Fig. 4a) by showing, in sharp contrast to the auditory modality, that the ability of participants to closely track the visual beat, vis. the quality of motor tracking, directly impairs performance accuracy. Moreover, this effect is selective for tempi presented close to natural motor dynamics (~1.7 Hz; Fig. 2b). In line with the auditory results, these analyses highlight that motor impact on temporal attention crucially depends on the temporal

simultaneity of motor acts relative to the beat, supporting a synergistic modulation of sensory processing that relies on the temporal alignment between motor and attention fluctuations. However, this does not explain why motor involvement positively impacts auditory temporal attention, but negatively impacts visual temporal attention.

**A model of coupled oscillators to generalise the results.** Finally, we investigated whether our results could be explained by a simple neural network model. To understand the specific motor contribution to auditory and visual periodic temporal attention, each having its own optimal sampling rate, we implemented a model in which sensory-specific temporal attention behaves like a self-sustained oscillator (a structure with an intrinsic rhythm capable of being entrained coupled to a motor oscillator and entrained by an external beat[51]. In its simplest realisation, this results in a model of three coupled phase oscillators (stimulus (S), attention (A) and motor (M) oscillators) with time-delays and noise[52] (Fig. 5a; see Methods section). The external stimulus (S) was a purely periodic rhythm (i.e., without distractors), which varied in frequency to mirror our different experimental conditions (between 0.3 and 3.8 Hz). The natural frequency of the sensory-specific oscillator (A) was fixed to reflect the optimal sampling rate of temporal attention, at 1.5 Hz for the auditory modality (after exp. 2; Fig. 2a) and at 0.7 Hz for the visual modality (after exp. 5; Fig. 4a). Finally, the natural frequency of the motor oscillator (M) was fixed at 1.7 Hz to reflect the spontaneous (free) tapping frequency (after exp. 3; Fig. 2b). Coupling strengths (K), time-delays ($\tau$) and the strength of the internal noise (D) were then adjusted to fit the different behavioural results (passive and tracking sessions in auditory and visual modalities; Fig. 5c, d). Behavioural performance was approximated by the phase-locking value (PLV[53] between the external beat (S) and the sensory-specific temporal attention oscillator (A), as it reflects the capacity of the sensory-specific oscillator to entrain to the external beat.

First, the model reproduced the results of the passive auditory (exp. 2; Fig. 2a) and visual (exp. 5; Fig. 4a) experiments (Fig. 5c, d). We approximated the results of these passive experiments with very high accuracy (auditory: fit quality: $R^2 = 0.92$, $p < 0.001$; visual: fit quality: $R^2 = 0.95$, $p < 0.001$). Importantly, apart from the natural frequency of the sensory-specific temporal attention oscillator (A; which differed between auditory (1.5 Hz) and visual (0.7 Hz) experiments) and the time-delay between the stimulus (S) and the motor (M) oscillator (auditory: $\tau_{S-M} = 0.1$ s; visual: $\tau_{S-M} = 0.35$ s) all other parameters (coupling strength K, time-delay $\tau$, and noise $\xi$) were similar across modalities ($K_{S-M} = 8$; $K_{S-A} = 10$, $\tau_{S-A} = 0.1$; $K_{A-M} = 10$, $\tau_{A-M} = 0$; $K_{M-A} = 2$, $\tau_{M-A} = 0$; $\xi_A = 5$; $\xi_M = 10$). Even if there is no explicit motor act in the passive session we assume that the motor system is already involved ($K_{M-A} = 2$), in line with a previous study[22]. Second, the model also successfully reproduced the results of the tracking auditory (exp. 2; Fig. 2a) and visual (exp. 5; Fig. 4a) experiments (auditory: fit quality: $R^2 = 0.95$, $p < 0.001$; visual: fit quality: $R^2 = 0.95$, $p < 0.001$), with a notably selective modulation of performance around 1.5–2 Hz in the tracking as compared to the passive sessions, which, crucially, was respectively positive and negative in the auditory and visual modalities. The only parameter that varied between passive and active sessions was the strength of the coupling between motor and temporal attention ($K_{M-A} = 10$; vs. 2 for the modelling of the passive sessions). Overall, three parameters played a key role in reproducing the behavioural results. In addition to the natural frequency of the sensory-specific temporal attention oscillator, which varied between modalities, and the coupling strength between motor

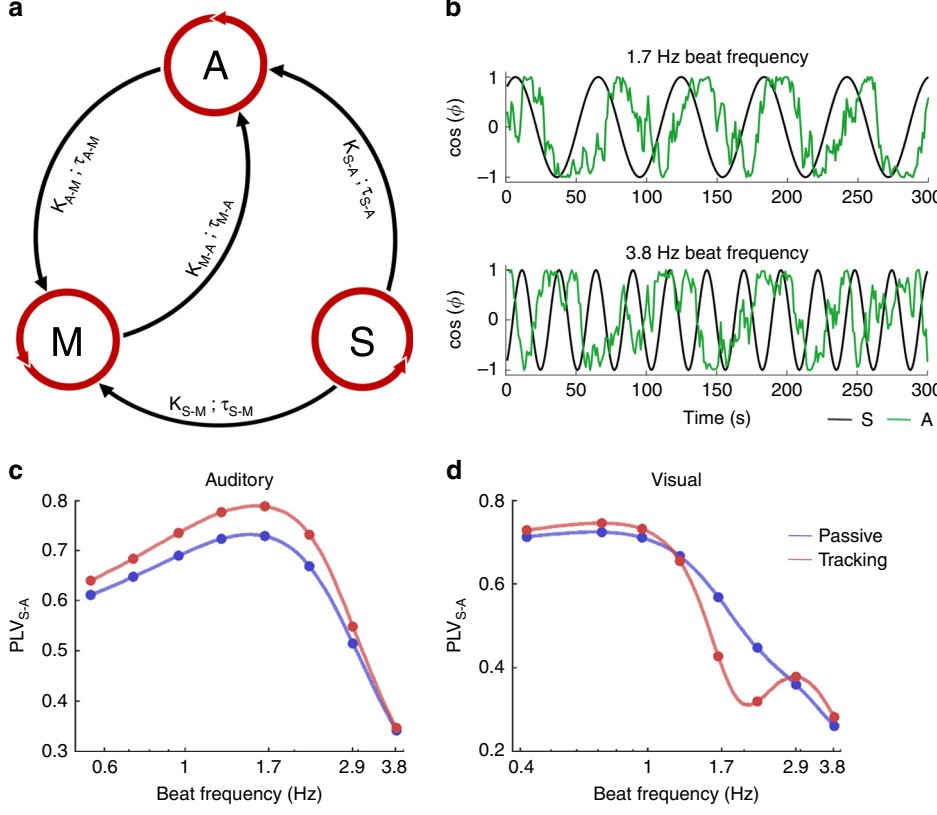

**Fig. 5 A model of coupled oscillators to generalise the results. a** Model of three delay-coupled phase oscillators approximating the selective coupling between the external beat (stimulus S), sensory-specific temporal attention (A), and natural motor dynamics (M). The external oscillator (S) influences attention and motor oscillators with a specific strength K and delay τ, and attention and motor oscillators reciprocally influence each other. The phase-locking value (PLV) between the external beat (S) and the sensory-specific temporal attention oscillator (A) reflects the capacity of A to entrain to S and thus is used as an approximation of behavioural performance. **b** Example of dynamics of the auditory temporal attention oscillator (A; green) during presentation of an external beat (S; black) at 1.7 Hz or 3.8 Hz. **c, d** Replication of the (**c**) auditory (exp. 2) and (**d**) visual (exp. 5) passive (blue) and tracking (red) experiments. Difference between conditions was obtained by adjusting three key parameters: the natural frequency of the sensory-specific temporal attention oscillator (A) and the time-delay between the stimulus (S) and the motor oscillator (M) which varied across modalities (auditory: $\omega_A = 1.5$ Hz, $\tau_{S-M} = 0.1$ s; visual: $\omega_A = 0.7$ Hz, $\tau_{S-M} = 0.35$ s), and the coupling strength between motor and attention oscillators, which varied between passive ($K_{M-A} = 2$) and tracking ($K_{M-A} = 10$) sessions.

and attention oscillators $K_{M-A}$, which varied between passive and tracking sessions, the time-delay between the stimulus (S) and the motor oscillator (M; $\tau_{S-M}$) was crucial for reproducing the difference of results across modalities.

## Discussion

Our findings reveal the natural sampling rate of periodic temporal attention. Attention supports the allocation of resources to relevant locations, objects, or moments in a scene[1]. Recent studies have revealed the rhythmic nature of sustained attention, showing that spatial[42,43,54–59] or featured-based[43] attention samples visual stimuli rhythmically, tethered by the phase of a theta (4–8 Hz) neural oscillation. Importantly, this temporal constraint is orthogonal to the attended (spatial or object-based) dimension, and hence does not hinder the quality of sensory selection. Here, in contrast, we reveal a surprisingly limited capacity—restricted to a lower range (0.5–2 Hz; Figs. 1c, 2a, 3c, 4a)—of humans to flexibly adapt and adjust their temporal attention to the natural dynamics of a scene.

On the one hand, these results give empirical support to studies investigating periodic temporal attention, which are consistently designed with rhythms in the 1–2 Hz frequency range[2,6–27,60]. On the other hand, they fuel recent frameworks postulating that the functional architecture of cognition is inherently rhythmic and

underpinned by neural oscillatory activity generated at the population level[61–64].

Confirming classic work focusing on motor synchronisation to periodic stimuli[29–31,49], our study shows that the temporal precision of motor acts is optimal when (auditory or visual) stimuli unfold a t around 1.7 Hz (Figs. 2c and 4b). But crucially, this set of experiments reveal the perceptual consequences of such sensorimotor synchronisation, by highlighting the intricate role of the motor system in temporal attention. The motor system is critically implicated in timing and time perception[65–68] and periodic—beat-based—timing, in particular, is underpinned by striato-thalamo-cortical circuits[65–68]. Our results confirm previous findings showing that overt motor activity optimises auditory periodic temporal attention[21,22]. They furthermore reveal that such an overt motor impact on temporal attention is rate-restricted and maximal around 1.7 Hz (Figs. 2a and 4a). This belongs to the delta (0.5–4 Hz) range of neural oscillations, which governs the dynamics of motor behaviour and motor neural processes[33]. Our findings also support previous results showing that motor delta oscillations represent temporal information and modulate perceptual processing[19,22,69–71]. Perception is thus shaped by motor activity, which unfolds at a delta rate and imposes temporal constraints on the sampling of sensory information. Strikingly, in our experiments, outside the range of natural motor dynamics overt movements had no significant impact on temporal attention,

either positive or negative. While participants were able to produce rhythmic movements between ~0.6 and 4.7 Hz (Fig. 2b), motor rhythmicity was less accurate outside ~1.7 Hz (Figs. 2b, c and 4b), and the inability of participants to closely track the beat—the lack of temporal simultaneity between motor acts and the beat—was associated with an absence of performance gain (Figs. 2f and 4e). Overt motor impact on temporal attention thus appears to be conditional upon the temporal alignment between motor and attention fluctuations.

An important question relates to the ubiquity of rhythmic sampling attentional mechanisms across modalities[61]. In this set of experiments, we directly applied the same paradigm, in two—auditory and visual—modalities. We observed in both of them, first, the existence of an optimal beat frequency at which temporal attention operates. Temporal attention is known to operate along the topographical dimension of the sensory system of interest[72,73] (spectral in audition, spatial in vision), and to synergistically interact with information present on such dimension to optimise perception[74,75]. Here, as sensory streams were providing clear and invariant spectral (in audition) or spatial (in vision) information, the impact of temporal attention on performance accuracy is thought to be optimal. Second, we showed that overt motor activity impacts temporal attention selectively for tempi presented close to the natural motor dynamics (~1.7 Hz; Fig. 2b), and furthermore highlighted that this effect crucially depends on the temporal simultaneity of motor acts relative to the beat. Nevertheless, several crucial differences across modalities exist. First, while auditory periodic temporal attention operates around 1.5 Hz, close to natural motor dynamics, visual periodic temporal attention operates around 0.7 Hz, that is, is twice slower (Fig. 3d). Our paradigm used transient stimuli which are known to be more suited to auditory than visual perception[76–78]. Indeed, the visual modality is ecologically more precise for capturing movement whereas audition is more adapted to transient stimuli[79]. Accordingly, participants were overall much more accurate in auditory than visual temporal attention (Fig. 3b)[77,80–84]. Specifically, fewer distractors per beat had to be inserted in the visual sequences to obtain similar levels of performance. However, independently of the overall accuracy effect, our results highlight a sensory-specific constrained sampling of temporal regularities, rather than an amodal optimal beat frequency at which temporal attention operates. These specific rhythmic sampling rates would thus emerge from the specific configuration of large-scale neural networks encompassing sensory (in addition to attentional and motor) regions[85,86]. Second, we reveal that the quality of motor tracking directly benefits performance accuracy in auditory attention, but negatively impacts it in visual attention (Figs. 2f and 4e). This was related to the fact that the temporal alignment between motor and attention fluctuations leading to optimal performance differed between modalities (Figs. 2f and 4e). In an active sensing framework, motor dynamics can act in concert with attention to temporally structure the activity of sensory cortices and shape perception[21,33,40,87]. Our results thus suggest that the delay for optimal coordination between motor and attentional fluctuations differs across modalities, with a better synchronisation in the auditory modality.

These two crucial differences between auditory and visual modalities were accurately captured in a model of coupled oscillators representing the periodic stimulus, a sensory-specific temporal attention oscillator and a motor oscillator (Fig. 5). Previous models of beat perception and temporal attention processes mainly focus on the capacity of a dynamical system composed of coupled oscillators tuned along a continuum of frequencies to temporally attend not only at a single frequency,

but also within a hierarchy of metric levels[23,24,44–46]. Here, we instead developed a model in which periodic temporal attention is conceived as one ongoing intrinsic oscillatory process with an optimal sampling rate. The difference of optimal sampling rate across modalities was directly related to the natural frequency of the sensory-specific temporal attention oscillator. More importantly, the time-delay between the stimulus and the motor oscillator was key in reproducing the differential impact of overt motor tracking on performance across modalities. While a small time-delay (100 ms) results in a positive motor impact on the quality of periodic temporal attention, a longer time-delay (350 ms) is associated with a disruptive effect. The presence of such a long delay in the visual modality is compatible with previous models of the visuomotor system[88]. Overall, this model captures the motor contribution to temporal attention in two sensory modalities. It reveals the structural constraints governing the temporal alignment between motor and attention fluctuations.

In conclusion, our results reveal the limited capacities of periodic temporal attention and its optimal sampling rate in two sensory modalities. They furthermore characterise the structural constraints governing the motor contribution to periodic temporal attention. Whether our results are specific to periodic temporal attention or generalise to other forms of temporal attending remains to be investigated[3,26,89–92].

## Methods

**Participants**. 30, 20, 50, 30, 20 and 15 participants (age range: 18–35 years; 69% of females) were respectively recruited for experiments 1 to 6. The experiments followed the local ethics guidelines from Aix-Marseille University. Informed consent was obtained from all participants before the experiments. All had normal audition and vision and reported no history of neurological or psychiatric disorders. We did not select participants based on musical training and a short survey made at the end of the experiment informed us that none of them were professional musicians.

**Experimental design of the auditory experiments (1, 2 and 6)**. Auditory stimuli were sampled at 44 100 Hz and presented binaurally at a comfortable hearing level via headphones (Sennheiser HD 250 linear) in an anechoic room, using the Psychophysics-3 toolbox[93] and additional custom scripts written for MATLAB (The Mathworks). Instructions were visually displayed on a mid-grey background on a screen laptop computer (Lenovo Thinkpad T470s) situated at a viewing distance of 50 cm. The screen had a spatial resolution of 1920 by 1080 pixels and vertical refresh rate of 60 Hz. On each trial, participants had to fixate a cross, located at the centre of the screen, to get a constant visual stimulation.

Each trial consisted of a sequence of pure tones, qualified as reference, targets and distractors (Fig. 1a). Three reference tones defining the beat of the sequence preceded a mixture of on-beat (target) and off-beat (distractor) tones. Participants performed a beat discrimination task at the end of each trial, by deciding whether the last tone of the sequence, a spectral deviant (785 Hz vs. 660 Hz), was on or off beat. While on-beat stimuli were providing the beat, crucially, off-beat stimuli had a distracting influence. This interleaved delivery of sensory events forced participants to track the beat throughout the entire duration of the sequence while minimising the interference of aperiodic events. This protocol thus ensured that their attentional focus was temporally modulated over an extended time period. This design implies that the last inter-stimulus interval (ISI) did not always correspond to the beat period of the trial. In particular, distractors could occur between the last target tone and the deviant. Tones frequencies were selected to avoid potential bones transmission (660/785 Hz). The beat frequency (or tempo) varied across conditions (8 conditions, with tempi of 0.6, 0.7, 1, 1.3, 1.7, 2.2, 2.9 and 3.8 Hz) to span the entire range of discernible beats[47,48]. In experiments 1 and 2, tones lasted 10% of the ISI (e.g., for a beat frequency of 1 Hz, the ISI would be of 1000 ms and the tones would last 100 ms). In experiment 6, we orthogonalized across conditions tempo and tone duration, by fixing across conditions tones length to 22.5 ms. Tones dampening length was of 10% of their duration and tones attenuation was of 40 dB. Trials had pseudo-random durations (~2–12 s) but included in each condition at least four targets (and up to 22; reference tones and deviant excluded) and lasted at least 2 s. These constraints were chosen to enable the deployment of temporal attention in all conditions. The density of distractors per sequence (i.e., the number of distractors per beat) was titrated individually (see below). Distractors appeared randomly between targets, with the constraint that all ISI within the sequence should be of at least 9% of the beat period (e.g., ISI > 90 ms for a 1 Hz beat). This ensured that tones were not appearing concomitantly. When the deviant was off beat, it appeared randomly within a window corresponding to one beat period and centred around the expected beat.

In all auditory experiments, participants performed a passive session, in which they executed the task while staying completely still during the duration of the trial, not moving any part of their body. In experiment 2, additionally, participants performed a 'tracking' session, in which they were required to follow the beat by moving their (left or right, at their convenience) index finger on a noiseless pad from the beginning of the sequence (the 2nd reference tone). The pad was home-made and included a microphone connected to a Focusrite Saffire Pro24 sound card to record participants movements. In essence, the tracking session is a variation of the synchronisation-continuation paradigm[31].

Each participant started the experiment with a short training session. The beat frequency was fixed to 2 Hz and the density of distractors was at first equal to zero and increased progressively up to 0.4 distractor per beat. Participants were instructed not to move during the trials (as in the passive session). Then, participants listened to the 8 conditions at least 1 time each. Following this short training session, participants performed a psychophysical staircase were the density of distractors was the varying parameter. The staircase was set to obtain ~75% of categorisation performance. Each experiment was divided into multiple sessions, each lasting around 1 h.

Participants performed 40 trials per condition per session. In experiments 1, 2 and 6, participants performed 2, 1 and 1 passive session, corresponding to 640, 320 and 320 trials, respectively. In experiment 2, they also performed 1 tracking session (320 trials). The order of the sessions, passive and tracking, was counterbalanced across participants. Conditions (tempi) were pseudo-randomly alternating in blocs of 20 trials each. Feedback was provided after each trial to indicate correct/incorrect responses, and more general performance feedback indicating the total number of correct responses was given after every bloc, for motivational purposes.

**Experimental design of the visual experiments (4 and 5).** These experiments are the transposition of experiments 1 and 2, respectively, to the visual modality. Each trial consisted of a sequence of centred visual gratings (visual extent 5°; Fig. 3a). Visual stimuli were sampled at 60 Hz. To impose a constant auditory stimulation on each trial, participants were presented with auditory pink noise binaurally at a comfortable hearing level via headphones. Participants performed a beat discrimination task at the end of each trial, by deciding whether the last grating of the sequence, a colour deviant (blue vs. yellow), was on or off beat. In experiment 4, 10 conditions were investigated, with beat frequencies of 0.3, 0.4, 0.6, 0.7, 1, 1.3, 1.7, 2.2, 2.9 and 3.8 Hz. Compared to the auditory experiments, two extra conditions (corresponding to tempi 0.3 and 0.4 Hz) were included, after pilot experiments, as it appeared that the optimal tempo is of lower range in the visual than auditory modality. In experiment 5, only 8 conditions were presented (for time constraints issues), corresponding to tempi of 0.4, 0.7, 1, 1.3, 1.7, 2.2, 2.9 and 3.8 Hz. Due to the presence of slower tempi, trials had pseudo-random durations of ~2–20 s. Gratings duration was longer than tones duration, to avoid presenting subliminal stimuli, and lasted 18% of the inter-stimulus interval (ISI; e.g., for a beat frequency of 1 Hz, the gratings would last 180 ms). Participants performed 40 trials per condition per session. In experiments 4 and 5, participants performed 3 and 1 passive session, corresponding to 880 and 320 trials, respectively. In experiment 5, they also performed 1 tracking session (320 trials).

**Free tapping experiment (3).** This experiment correspond to a subset of BAASTA (Battery for the Assessment of Auditory Sensorimotor and Timing Abilities)[49]. To assess participants' spontaneous tapping rate and motor variability without a pacing stimulus, participants were asked to tap regularly at a comfortable rate for 60 s, the only instruction being to maintain the tapping rate as constant as possible. In two additional conditions, participants were instructed to tap rhythmically as fast and as slow as possible, for 30 and 60 s, respectively. Participants were required to tap with their index finger on the noiseless home-made pad.

**Timing of motor acts in tracking sessions.** To investigate the ability of participants to actively follow the beat, we extracted the timing of individual motor acts in the tracking sessions. For each trial, we computed the mean and standard deviation of the inter-tap intervals. We then derived the guided tapping precision, expressed as the ratio (in %) between the average tapping frequency and the tempo, and the coefficient of variation (CV), expressed as the relative standard deviation, i.e., the ratio between the standard deviation of the inter-tap intervals and the tempo. For each trial, we also estimated the temporal distance between each individual motor acts and the beat, which indexes the sensorimotor simultaneity, i.e., the degree of simultaneity of motor acts relative to on-beat stimuli. This temporal distance was then normalised relative to the beat period (in a $2\pi$ space), with zero corresponding to perfect simultaneity between a motor act and the beat. We then either derived a relative or absolute sensorimotor simultaneity index, which respectively allows to estimate if participants tended to tap in anticipation or reaction to the beat, or to quantify the degree of sensorimotor simultaneity.

**Estimation of an optimal beat frequency.** To estimate the tempo at which performance (/CV) would be maximal (/minimal), variations of performance (/CV) across conditions were approximated with a third order polynomial function

$f(x) = ax^3 + bx^2 + cx + d$, and the coordinates of the local maxima α (/minima β) were extracted according to the functions:

$$\alpha = \frac{-b - \sqrt{\delta}}{3a}; \beta = \frac{-b + \sqrt{\delta}}{3a}$$

where $\delta = b^2 - 3ac$ with $\delta > 0$. We used a third order polynomial function, as it is the best (ie most flexible) model that allows estimating one maximum without ambiguity (higher order models accept multiple maxima).

**Statistical procedures.** All analyses were performed at the single-subject level and followed by standard parametric two-sided tests at the group level (repeated-measures ANOVAs, paired and unpaired $t$-tests, Spearman correlations). For unpaired $t$-tests, we used two by two $t$-test Welsh correction. When necessary, to provide an unbiased decision criterion with regards to the null hypothesis, we additionally used Bayesian statistics to derive a Bayes factor. We used a standard approach to compute the Bayes factors between "null" and "effect" hypotheses at the population level using the Akaike Information Criterion[94]. According to this symmetric hypothesis comparison measure, a Bayes factor of <1/3 provides significant evidence in favour of the null hypothesis. Bayes factors were also computed for correlation coefficients[95].

**Generalised linear mixed model.** A generalised linear mixed model (GLMM) regression analysis was performed in R (glmer function) on the passive auditory experiment (exp. 1) to characterise the extent to which performance was impacted by the nature of the deviant (on-beat or off-beat), its distance relative to the beat (in ms, in relative value) and the beat frequency. To do so, performance was analysed with these three variables used as predictors and participant number used as random factor.

**Model of coupled oscillators.** We implemented a model of three coupled phase oscillators[96] with time-delays and noise[52] to approximate the selective coupling between a periodic external beat (stimulus; S), sensory-specific periodic temporal attention (A) and motor tapping (M) (Fig. 5a). The model was implemented with a set of differential equations, as:

$$\theta_S = \omega_S,$$

$$\theta_A = \omega_A + K_{AM}\sin[\theta_M(t - \tau_{AM}) - \theta_A] + K_{AS}\sin[\theta_S(t - \tau_{AS}) - \theta_A] + \xi_A(t),$$

$$\theta_M = \omega_M + K_{MA}\sin[\theta_A(t - \tau_{MA}) - \theta_M] + K_{MS}\sin[\theta_S(t - \tau_{MS}) - \theta_M] + \xi_M(t),$$

where $\omega_i$, $\theta_i$ and $\xi_i$ are the natural frequency, phase and noise of an oscillator i, and for each pair of oscillators i and j, $K_{ij}$ and $\tau_{ij}$ represent the coupling strength and time-delay from oscillator i to j. The noise $\xi_i$ is additive and Gaussian with an intensity D, such as $\langle \xi_i(t) \rangle = 0$ and $\langle \xi_i(t)\xi_j(t') \rangle = 2D_{ij}\delta(t - t')\delta_{ij}$ (where $\langle \cdot \rangle$ denotes the time-average operator and $\delta$ the delta function). In line with studies implementing models of coupled oscillators, we ran the simulation during 1e4 seconds, in order to obtain an equilibrium in the interaction between the coupled oscillators[52,97] and to approximate the duration (summed across trials and participants) of the behavioural experiments. The sampling rate of the simulation was 25 ms. We thus set internal time-delays to 0 ms (i.e., <25 ms).

The level of coherence between the external beat S and the sensory-specific oscillator A was computed with the phase-locking value (PLV)[52,53]. It estimates the capacity of A to entrain to S and is hence used as an approximation of behavioural performance. PLV is defined as:

$$\text{PLV}_{S-A} = \frac{1}{N}\sum_{t=1}^{N} e^{i\Delta\theta_{SA}(t)},$$

where the phase angle $\Delta\theta$ between oscillators S and A at time t is averaged across time points from $t = 1$ to N.

**Reporting summary.** Further information on research design is available in the Nature Research Reporting Summary linked to this article.

## Data availability

The entire dataset of this study is available on GitHub: https://github.com/DCP-INS/TempAtt.

## Code availability

Codes to reproduce the results and figures of this manuscript are available on GitHub: https://github.com/DCP-INS/TempAtt. Codes to run the experiments are available from the corresponding author upon reasonable request.

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

## Acknowledgements

We thank Viktor Jirsa and Jennifer Coull for their valuable insights into this paper. Research supported by grants ANR-16-CONV-0002 (ILCB), ANR-11-LABX-0036 (BLRI) and the Excellence Initiative of Aix-Marseille University (A*MIDEX). S.P. had funding from the European Union's Horizon 2020 Framework Programme for Research and Innovation under the Specific Grant Agreement no. 785907 (HBP SGA2).

## Author contributions

A.Z. and B.M. designed the experiments; A.Z. acquired data; A.Z. and B.M. analysed data; S.P. implemented the model; and A.Z. and B.M. wrote the paper.

## Competing interests

The authors declare no competing interests.
