## [Peer Review File · Nature Communications]

Reviewers' Comments:

Reviewer #1:

Remarks to the Author:

Utilizing six experiments and a simulation of a simple three-coupled-oscillator system, the authors examined performance on auditory and visual temporal judgment tasks, both with and without a concurrent requirement to tap along isochronously with the tactus underlying the sequence of events. Across blocks of trials, the beat frequency was varied. In both modalities, performance across beat frequencies was best described by an inverted U-shaped function, with the average optimal beat frequency for auditory sequences being approximately twice as fast as for visual sequences. The optimal beat frequency for auditory sequences was closely matched to the spontaneous tapping rate of the participants. In a range of beat frequencies close to the spontaneous motor tempo, tapping along with auditory sequences facilitated performance on the temporal judgment, whereas it interfered with performance on the visual trials. The pattern of behavioral results was modeled quite well with a system of coupled oscillators representing stimulus, motor, and attentional periodicities.

These results contribute to the recent literature, much of it from the senior author's lab, about the properties of (periodic) temporal attention, and they build on decades of research on temporal attention (primarily in the auditory realm). The experiments and results nicely illustrate the presence of distinct intrinsic periodic processes that facilitate auditory and visual perceptual processing. The observation that auditory sensorimotor coupling facilitates perceptual performance whereas visual sensorimotor coupling disrupts perceptual performance is perhaps the most intriguing new finding.

My main concerns with the manuscript have to do with the framing and terminology, and also the lack of important methodological detail. I elaborate on these concerns below.

Comments and concerns pertaining to theoretical framing and terminology:

The conceptualization of temporal attention is at times loose and misses some highly relevant previous literature.

First, there is some ambiguity with regard to temporal attention (the capacity to attend to specific moments in time) more generally and periodic temporal attention more specifically. They allude to this difference in the last sentence of their paper when they write, "Whether our results are specific to periodic temporal attention or generalise to other forms of temporal attending remains to be investigated 3,19,84–87." Having thus appropriately narrowed the scope of the phenomenon they are examining, it seems inappropriate to invoke the more general concept of temporal attending as they do in the penultimate sentence, "They furthermore characterize the structural constraints governing the motor contribution to temporal attention." At a minimum, I believe the authors should explicitly write "periodic temporal attention," or PTA as an acronym, unless they are explicitly discussing a more general form of temporal attention.

Even within the narrower conception of periodic temporal attention that invokes the concept of endogenous oscillators that are tuned in a modality-specific manner, the authors fail to cite a substantial body of well-known work that has specifically addressed such issues. The most notable omission is the paper of Large & Jones (1999). Also relevant is Large's more recent modeling work pertaining to beat extraction that involves layers of oscillators, where the layers are conceived of as sensory and motor layers (Large et al., 2015).

Oscillator models have also been used recently to model temporal attention and successfully predict performance in psychophysical tasks using musical rhythms that are more complex than isochronous rhythms (Hurley et al., 2018).

The title is misleading. First it implies that participants attended to rhythms, (musical) patterns of durations, presumably in order to make judgments pertaining to said rhythms. Although (irregular) rhythms were created by the insertion of distractors, participants weren't actually asked to attend to rhythms or discriminate among rhythms. Instead, participants had to listen for the occurrence in the change in frequency of the last-played tone in order to make a on/off-beat judgment about that tone. That's not the same as attending to rhythms.

The word "beat" is used incorrectly throughout most of the paper. Most of the time, when the authors refer to "beat" in the vernacular, they are actually referring to the beat frequency, or tempo of the sequence. In writings dealing with concepts of tempo, meter, and rhythm, the word "beat" is typically used to refer to one of the individual moments in time that coincide with the tactus of the sequence.

The framing, in the Abstract, that an underlying rhythmic process might hamper temporal attention is interesting, but it isn't clear what it refers to. Does it refer to the interference of the motor rhythmic process on the perception of visual sequences, or is it making a more general claim that rhythmic processes might hamper temporal attention? If the latter, then the formulation seems a bit odd because one might think of the underlying periodic process as facilitating attention within some range of periodicities, rather than inhibiting it. This should be clarified.

pg. 15 "On the one hand, these results retrospectively explain why studies investigating periodic temporal attention are consistently designed with rhythms in the 1-2 Hz frequency range 2,6–27" One could also argue, that because it is obvious that attention is going to operate on sequences of events whose rates of onsets fall into this range, e.g. language and music, previous research on temporal attention settled precisely into this range of stimulation rates on purpose rather than by accident. Sensory and motor phenomena implemented in neural systems are (necessarily) constrained to exist within appropriate frequency ranges, so the existence of such constraints on mechanisms that link sensory and motor systems is not news, per se. One example of previous research on this topic is that of Drake et al (2000) in which both perceptual and motoric constraints on synchronization behaviors are described (as a function of age) in terms of attentional oscillators. Demonstrating modality-specific sensory-motor interactions in periodic attention, as the authors do, is more significant.

"Through six interrelated behavioural experiments, we reveal the existence of a limited sampling capacity of temporal attention, which is moreover sensory-specific." Given the structure of the tasks, i.e. the manipulation of distractor density in order to achieve certain levels of performance, one could argue that the paper is actually examining the ability of the brain to entrain an oscillator to a periodicity that might then guide attention, rather than an ongoing oscillator that is sampling its sensory environment, irrespective of what the sensory input is doing. It isn't clear to me how the experiments distinguish these two possibilities, and, as noted below, there is reason to believe that, at least for the auditory stimuli, the arrhythmic properties of the sequences created by the distractors are hampering the entrainment of a periodic attention process, especially if the periodicity is farther away from the preferred frequency. It is unclear whether the "sensory" input to the model also contained the distractors. It seems that the best way to model the behavior would be to feed into the model, on a trial-by-trial basis, the stimuli that participants actually heard.

Methodological concerns:

Several methodological details about the construction of the trials are lacking:

First, it is unclear what the variability was in the number of possible events in the sequence. It is written that, "trials had pseudo-random durations (~2 to 10s) but included at least four targets and lasted at least 2 seconds." Given the different tempi, different trial durations would be expected for

fixed numbers of events as a result, but the implication of the authors' statement is that the number of events varied also. What was the range of the number of possible events, and was this range the same for all tempi? Were the three reference tones taken into account when calculating the range of reported trial durations?

Second, on pg. 18, the magnitude(s) of the temporal deviation of the last event about which an on/off-the-beat judgment had to be made is not specified, but is absolutely critical to know. Was it a fixed amount or a proportion of the beat-frequency? If it was a proportion, then depending on tempo there could be an issue of temporal discrimination limits at higher rates (having nothing to do with attention). Across trials, could off-beat events either precede or follow the expected onset time, or were they consistently either early or late? Figure 1a implies that there was a range of off-beat times both before and after the expected event time, but this needs to be described clearly in the text.

Third, what proportion of trials were "on-beat" trials? Related to this is the question of why the authors are reporting proportion correct rather than the signal detection theory measure, d' , as would be more typical.

Fourth, did distractors occur between the last target event and the to-be-judged event?

Fifth, Figure 3b indicates that the majority of participants had a "difficulty" level set to fewer than 1 distractor per beat, which means that there were many targets between which no distractor would occur (particularly for vision). What therefore determined whether a distractor would be placed between any given pair of targets? Figure 3a shows a highly atypical sequence, in that there is at least one distractor between each pair of targets. However, only one participant attained a performance level of 1 distractor/beat in the visual condition. Please show a more representative trial.

Sixth, I think the answer is no, but please state explicitly whether deviants were constrained to fall on a metric grid? Given that the "target" and "distractor" tones had the same frequency (660 Hz?), non-metric distractors would be expected to have a *profound* impact on the ability to entrain to the tactus. What this issue highlights is the fact that this paper is more about the ability of aperiodic events to disrupt an endogenous oscillation than it is about the ability of an endogenous oscillation to facilitate perception. Previous studies of temporal attention have manipulated the timing of the target about which a judgment is made in order to determine how temporally constrained the focus of attention is, e.g. Barnes & Jones (2000); Jones et al. (2002). The fact that a different aspect of periodic temporal attention (its resilience to distractors) is being examined in this paper isn't really made clear.

Relatedly, almost 50 years of auditory scene analysis research suggests that the frequency separation of the "target" and "distractor" tones should have a profound effect on the ability to maintain attention at the beat frequency, and therefore on the results. If the metronomic "targets" were to segregate into their own stream, and participants can attend to that stream, would a similar profile across beat frequencies be observed? I'm not sure it would. This isn't to say that there wouldn't still be a preferred beat frequency (tempo), but if all of a sudden performance were to increase at non-optimal rates, e.g. 2.9 Hz, what would that have to say about periodic attention at a single preferred rate?

pg. 19. "Following this short training session, participants performed a psychophysical staircase where the density of distractors was the varying parameter. The staircase was set to obtain 75 % of categorization performance." Was the distractor density value obtained for 2 Hz applied to all of the other beat frequency conditions? If it was set based on 2 Hz, should the curves in Figures 1c and 3c not be passing through 0.75? It appears that it is pretty close in 1c, but not in 3c.

Minor points:

pg. 3 "A succession of events appearing too fast, or two stimuli too close in time, are situations typically difficult to attend to." This seems to be conflating the ability to discriminate individual events, attending to individual events, and attending to a stream of events. For example, it is easy to attend to a roll performed on a snare drum, even though one may not be focusing attention on the sound made by each hit on the drum.

pg. 5 "The comparison of conditions revealed significant fluctuations in performance (% correct responses) across beats (repeated-measures ANOVA: $F(7,196) = 17.6, p < .001$; Fig. 1c)." Presumably by "across beats" you mean "across tempi" or "across beat frequencies" as opposed to individual beats within a trial.

pg. 5 "They moreover had an inverse U-shape profile" Unclear what "they" is referring to. Individual subject performance profiles across tempi?

pg. 6 "These results suggest that during auditory perception temporal attention presents an optimal sampling rate, around 1.2 Hz." Very similar to the sentence two sentences before this one. Not sure what is gained by restating it.

pg. 6 "we thus orthogonalized beats and stimulus duration" I am assuming that by "beats" you again mean tempi. Because the meaning of the word beat is ambiguous, I encourage the authors to use the more conventional term, tempo, or beat frequency to refer to the different rates at which individual beat-keeping events are presented.

pg. 13 Reference to Fig. 6b-c is probably supposed to be Fig. 5b-c.

pg. 16 "Accordingly, participants were overall much more accurate in auditory than visual temporal attention (Fig. 3b)" It's unclear how Figure 3b shows this. Figures 1c and 3c would suggest that, if anything, participants were more accurate in the visual conditions.

References:

R. Barnes and M. R. Jones (2000). "Expectancy, attention, and time." *Cognitive Psychology* 41(3): 254-311.

C. Drake, M. R. Jones and C. Baruch (2000). "The development of rhythmic attending in auditory sequences: attunement, referent period, focal attending." *Cognition* 77(3): 251-288.

B. K. Hurley, L. K. Fink and P. Janata (2018). "Mapping the dynamic allocation of temporal attention in musical patterns." *Journal of Experimental Psychology: Human Perception and Performance* 44(11): 1694-1711. <http://doi.org/10.1037/xhp0000563>

M. R. Jones, H. Moynihan, N. MacKenzie and J. Puente (2002). "Temporal aspects of stimulus-driven attending in dynamic arrays." *Psychological Science* 13(4): 313-9.

E. W. Large, J. A. Herrera and M. J. Velasco (2015). "Neural Networks for Beat Perception in Musical Rhythm." *Frontiers in Systems Neuroscience* 9(159). <http://doi.org/10.3389/fnsys.2015.00159>

E. W. Large and M. R. Jones (1999). "The dynamics of attending: How people track time-varying events." *Psychological Review* 106(1): 119-159.

Reviewer #2:

Remarks to the Author:

This paper reports a series of six behavioral psychophysical experiments on auditory and visual temporal attention using a beat discrimination task. The results suggest that optimal sampling rates are higher in audition than in vision and that a secondary motor tracking task facilitates auditory temporal discrimination but impairs visual discrimination. The authors present a simple neural network model that can fit the data well.

Evaluation

Comparing the psychophysics of auditory vs. visual perceptual processing is useful, and the finding of opposite effects of a secondary motor tracking (i.e., tapping) task on vision vs. audition is possibly the most interesting result of this study. At this stage I am still somewhat ambivalent in my evaluation of the contribution of this paper. In my view, the task-specificity of modality-specific effects would need to be highlighted more, and there are some aspects of the design that limit a strict comparability of performance across the senses.

Comments

According to the notion of modality appropriateness, audition is specialized for temporal processing, whereas vision is specialized for spatial processing. Hence, when examining temporal discrimination processes, audition should be superior to vision, whereas when examining spatial discrimination it should be the reverse data pattern. This has been shown, repeatedly, in studies on crossmodal visual-auditory spatial attention and also crossmodal temporal attention. In my view, this issue of task-specificity of modality-specific effects need to be discussed in more detail. Moreover, the finding of sensory visual impairment when coupled with motor tapping would be more impressive if the reverse "dissociation" could be demonstrated as well, that is, impaired auditory spatial discrimination performance whilst visual spatial attention is improved. Currently, we have a single finding, and it is hard to know whether this tells us about sensory systems, task characteristics, or the interaction of both (I suspect the latter). Hence, while I believe that the present data are very interesting, the potential contribution could be further strengthened by discussion of (and possibly data on) the issue of whether the task demands spatial vs. temporal discrimination. Currently, the authors do not offer a strong theoretical account as to why this specific interaction is observed. (The fact that a neural network model can fit the data well shows us that the data are reasonably systematic but does not help us too much to "explain" the data pattern from a cognitive point of view.. Such a discussion with respect to how sensory codes and motor codes interact in a domain- and task-specific way would also help to evaluate the novelty value of the present set of results.

My impression is that the comparability of performance across the senses needs even more discussion. One aspect to be discussed is the individual titration of presentation rates, and possibly another would be the (possibility of) demonstration of psychophysical equivalence in terms of perceived stimulus intensity, etc. I am not meaning this "overcritically", but I am just mentioning that it is really difficult to interpret "simple interaction" effects across modality. Perhaps this could be mentioned somewhere.

The authors argue that flexibly focusing in time and its limits has "never been investigated", but this is probably overstated because there are studies on global vs. local focus of auditory attention (e.g.,

work by List or by Sanders, etc.) and even on switching the attentional focus between global and local auditory patterns. This could be mentioned (and the statement could be softened or specified accordingly).

p. 6, line 139: Being no methods expert, I am still wondering whether the dfe for paired welsh t test can be 38 if it refers to a study with $n = 20$? At least this makes me wonder what experiment and which data is being analyzed here and which are the two conditions to be compared. Perhaps a little rewriting could help the reader (or at least this particular reader) to be able to follow better the description of the results.

p. 7, line 165: a BF of .32 and .33 closely corresponds to $1/3$, so is this taken as evidence for the H_0 ? A verbal paraphrase would help here in order to know what the authors would like to conclude from such statistics.

p. 8, line 169: The headline in bold makes a statement that seems to be contradicted by the last sentence of the same paragraph. Perhaps this could be clarified.

p. 9, line 217: Detail: did not revealed -> reveal

p. 10, line 238: $t(14)$ -> please describe the conditions/participants included in this comparison.

p. 11, line 276; theses analyses -> these

Reviewer #3:

Remarks to the Author:

Review of "Attention to rhythms: sensory-specific constrained sampling of temporal regularities" by Zalta, Petkoski, & Morillon

In six experiments the authors investigate the rhythmic behavior of temporal attention in three different modalities (visual, auditory, and motor). The paradigm they employed consisted of a sequence of stimuli presented at different frequencies. After the first three stimuli, additional distractor stimuli were presented at various temporal proximities. The participants' task was to determine whether the very last stimulus was either on or off-beat in comparison with the first three reference stimuli. By systematically varying the frequency of the stimuli they were able to determine that the most optimal beat frequency was about 1.4 Hz for auditory stimuli, 0.7 Hz for visual, and 1.7 Hz for motor stimuli, reflecting the modality-specific sampling rates of sensory perception and temporal attention. In addition, they revealed that finger tapping along with the presented rhythm actually helped with auditory performance when it was between 1.3 and 2.2 Hz, thus close to the optimal/preferred frequency of both audition and motor control. In contrast, tapping along with visual stimuli had no effect on performance except at 1.66 and 2.2 Hz where it actually interfered with visual performance. Finally, the authors present a computational model consisting of three delay-coupled phase oscillators to explain the pattern of findings.

General evaluation: I really enjoyed reading this well-written paper, which presents highly interesting results regarding the rhythmic behavior of temporal attention within and across different modalities, confirming the modality-specific nature of attentional restrictions. The experiments are elegantly designed and the analyses and interpretation of the results are well done.

I only have a couple of questions/suggestions as detailed below:

Experiment 1 and 2 provided somewhat different estimates of the optimal frequency for audition (1.2 hz in exp 1 and 1.5 hz in exp 2). I might have missed it, but a discussion why this might be the case seems to be missing. A potential explanation might lie in individual/group differences, and for that reason it is a pity that the authors did not control or at least measure the amount of musical experience that their participants had. It is not hard to imagine that a trained musician must show a lot less variability in tapping behavior as well as how it interacts with temporal attention in the visual modality (which might interfere even more) and the auditory modality (which might be more beneficial).

Line 203: Please rephrase/elaborate this sentence, as I initially got confused by the "The average difficulty level ...was significantly lower than in the auditory task". That is, it took me some time to realize that the visual task was actually much harder than the auditory task and that the difficulty level thus needed to be set lower for the visual task (introducing fewer distractors) to obtain comparable levels of performance.

In line 230 the authors argue that participants tended to tap too fast in all conditions of the visual task. One page later though (line 258) it is written that participants were not anticipating the beat but tapped in reaction to it. Please explain.

In lines 276-280, it is argued that a synergistic modulation of sensory processing seems to rely on the temporal alignment between motor and attention fluctuations (i.e. 1.5 hz (aud) being closer than 1.7 hz (motor) than 0.7 (vision) to 1.7 hz), but that this does not explain why motor involvement positively impacts auditory but not visual attention. Do I understand correctly that the time-delay between stimulus and motor oscillator was indeed the crucial additional component in your model to explain the pattern of results?

It is a pity that the authors did not look at the interactions between the visual and auditory modalities as well by having some conditions within rather than only between participants. It would be interesting to see whether an individual who performs well in the auditory task would perform either better or worse in the visual task. Would it be possible to make any predictions using the existing model?

I miss information as to whether participants were able to see their own (tapping) finger during the tasks. Is it possible that the movement of the finger could have had a distracting influence on performance (more so in the visual than in the auditory task)?

I'm surprised that no further instruction/restriction was needed on how slow participants were allowed to tap in the 'as-slow-as-possible-free-tapping-task'.

Figure 1 (and 3): It would be helpful to explicitly/visually indicate the reference stimuli, the mixture of target and distractor stimuli, and the final target stimulus.

Line 717: It is rather hard to recognize the dashed line in panel C, i.e., it is not clear that it is dashed.

Figure 2b: Figure seem to be cut off at the right edge.

Line 758: How/where was threshold performance defined?

Signed, Sander Martens

Response to reviewers

We would like to thank the reviewers and the editor for their constructive and positive evaluation of the manuscript. In the following, we reply to the reviewers' comments in a point-by-point manner: Plain font: unabridged reviewer comments; Blue font: our reply. Changes in the manuscript are also highlighted in blue.

Reviewers' comments:

Reviewer #1 (Remarks to the Author):

Utilizing six experiments and a simulation of a simple three-coupled-oscillator system, the authors examined performance on auditory and visual temporal judgment tasks, both with and without a concurrent requirement to tap along isochronously with the tactus underlying the sequence of events. Across blocks of trials, the beat frequency was varied. In both modalities, performance across beat frequencies was best described by an inverted U-shaped function, with the average optimal beat frequency for auditory sequences being approximately twice as fast as for visual sequences. The optimal beat frequency for auditory sequences was closely matched to the spontaneous tapping rate of the participants. In a range of beat frequencies close to the spontaneous motor tempo, tapping along with auditory sequences facilitated performance on the temporal judgment, whereas it interfered with performance on the visual trials. The pattern of behavioral results was modeled quite well with a system of coupled oscillators representing stimulus, motor, and attentional periodicities.

These results contribute to the recent literature, much of it from the senior author's lab, about the properties of (periodic) temporal attention, and they build on decades of research on temporal attention (primarily in the auditory realm). The experiments and results nicely illustrate the presence of distinct intrinsic periodic processes that facilitate auditory and visual perceptual processing. The observation that auditory sensorimotor coupling facilitates perceptual performance whereas visual sensorimotor coupling disrupts perceptual performance is perhaps the most intriguing new finding.

My main concerns with the manuscript have to do with the framing and terminology, and also the lack of important methodological detail. I elaborate on these concerns below.

We thank the reviewer for this positive and detailed review of our work. In particular, for taking the time to help us having a precisely framed manuscript and for pointing out the missing methodological details and relevant references. We believe our revised manuscript is now much clearer and precise.

Comments and concerns pertaining to theoretical framing and terminology:

The conceptualization of temporal attention is at times loose and misses some highly relevant previous literature.

First, there is some ambiguity with regard to temporal attention (the capacity to attend to specific moments in time) more generally and periodic temporal attention more specifically. They allude to this difference in the last sentence of their paper when they write, "Whether

our results are specific to periodic temporal attention or generalise to other forms of temporal attending remains to be investigated 3,19,84–87." Having thus appropriately narrowed the scope of the phenomenon they are examining, it seems inappropriate to invoke the more general concept of temporal attending as they do in the penultimate sentence, "They furthermore characterize the structural constraints governing the motor contribution to temporal attention." At a minimum, I believe the authors should explicitly write "periodic temporal attention," or PTA as an acronym, unless they are explicitly discussing a more general form of temporal attention.

Thank you for pointing out this inconsistency in our manuscript. We were indeed referring, in the penultimate sentence, to the specific form of 'periodic temporal attention'. We thus added the term 'periodic' to the sentence and verified that throughout the manuscript that the distinction between the specific (periodic) and general forms of attention is clearly explained. Please notice that we were already discussing the different forms of temporal attention in the introduction section:

p. 3: "Multiple types of temporal structures are capable of guiding temporal attention ¹, such as isochronous ² or heterochronous streams of events ³, symbolic cues ⁴, or hazard functions ⁵. Paradigms that involve isochronous perceptual streams [...]"

Even within the narrower conception of periodic temporal attention that invokes the concept of endogenous oscillators that are tuned in a modality-specific manner, the authors fail to cite a substantial body of well-known work that has specifically addressed such issues. The most notable omission is the paper of Large & Jones (1999). Also relevant is Large's more recent modeling work pertaining to beat extraction that involves layers of oscillators, where the layers are conceived of as sensory and motor layers (Large et al., 2015).

Oscillator models have also been used recently to model temporal attention and successfully predict performance in psychophysical tasks using musical rhythms that are more complex than isochronous rhythms (Hurley et al., 2018).

We apologize for the fact that this literature was missing in our manuscript. It was actually supposed to be mentioned, because it obviously inspired this work.

First, we implemented the following sentence in the introduction to clarify why we choose an oscillatory model:

p. 4: "Finally, in line with previous models of beat perception and temporal attention processes 15,16,45–47, we show that our results are reproduced by a simple model involving three coupled oscillators."

We moreover now refer in the manuscript to the following articles:

15. Jones, M. R. Time, our lost dimension: toward a new theory of perception, attention, and memory. *Psychol. Rev.* **83**, 323–355 (1976).
16. Jones, M. R., Moynihan, H., MacKenzie, N. & Puente, J. Temporal aspects of stimulus-driven attending in dynamic arrays. *Psychol. Sci.* **13**, 313–319 (2002).
45. Large, E. W. & Jones, M. R. The dynamics of attending: How people track time-varying events. *Psychological Review* **106**, 119–159 (1999).
46. Large, E. W., Herrera, J. A. & Velasco, M. J. Neural networks for beat perception in musical rhythm. *Front. Syst.*

Neurosci. **9**, 1–14 (2015).

47. Hurley, B. K., Fink, L. K. & Janata, P. Mapping the dynamic allocation of temporal attention in musical patterns. *J. Exp. Psychol. Hum. Percept. Perform.* **44**, 1694–1711 (2018).

The title is misleading. First it implies that participants attended to rhythms, (musical) patterns of durations, presumably in order to make judgments pertaining to said rhythms. Although (irregular) rhythms were created by the insertion of distractors, participants weren't actually asked to attend to rhythms or discriminate among rhythms. Instead, participants had to listen for the occurrence in the change in frequency of the last-played tone in order to make a on/off-beat judgment about that tone. That's not the same as attending to rhythms.

We agree that the beginning of the title was misleading, and propose this version:

"Natural rhythms of periodic temporal attention"

The word "beat" is used incorrectly throughout most of the paper. Most of the time, when the authors refer to "beat" in the vernacular, they are actually referring to the beat frequency, or tempo of the sequence. In writings dealing with concepts of tempo, meter, and rhythm, the word "beat" is typically used to refer to one of the individual moments in time that coincide with the tactus of the sequence.

Thank you for pointing out this imprecision. We replaced the term "beat" by "tempo(i)" or "beat frequency" throughout the manuscript, when referring to the frequency of the sequence.

The framing, in the Abstract, that an underlying rhythmic process might hamper temporal attention is interesting, but it isn't clear what it refers to. Does it refer to the interference of the motor rhythmic process on the perception of visual sequences, or is it making a more general claim that rhythmic processes might hamper temporal attention? If the latter, then the formulation seems a bit odd because one might think of the underlying periodic process as facilitating attention within some range of periodicities, rather than inhibiting it. This should be clarified.

Indeed, the term "hampered" was ambivalent. We had the latter description mentioned by the reviewer in mind. We now state:

p. 2: "Whether this function is bounded by an underlying rhythmic neural mechanism is unknown".

pg. 15 "On the one hand, these results retrospectively explain why studies investigating periodic temporal attention are consistently designed with rhythms in the 1-2 Hz frequency range 2,6–27" One could also argue, that because it is obvious that attention is going to operate on sequences of events whose rates of onsets fall into this range, e.g. language and music, previous research on temporal attention settled precisely into this range of stimulation rates on purpose rather than by accident. Sensory and motor phenomena

implemented in neural systems are (necessarily) constrained to exist within appropriate frequency ranges, so the existence of such constraints on mechanisms that link sensory and motor systems is not news, per se. One example of previous research on this topic is that of Drake et al (2000) in which both perceptual and motoric constraints on synchronization behaviors are described (as a function of age) in terms of attentional oscillators. Demonstrating modality-specific sensory-motor interactions in periodic attention, as the authors do, is more significant.

We thank the reviewer for pointing out this ambivalent formulation. Even if previous work had used such range on purpose, the present results provide critical evidence that periodic temporal attention optimally operates in the 0.5-2 Hz frequency range. To more precisely highlight this point, we modified the sentence cited above by the following one:

p.14: "On the one hand, these results give empirical support to studies investigating periodic temporal attention which are consistently designed with rhythms in the 1-2 Hz frequency range."

We also now cite the article mentioned by the reviewer in the discussion section.

63. Drake, C., Jones, M. R. & Baruch, C. The development of rhythmic attending in auditory sequences: Attunement, referent period, focal attending. Cognition 77, (2000).

Nevertheless, we do not think that it was obvious that sensory phenomena are constrained at specific frequency ranges. For example, speech rate is measured around 4.5 Hz which is far away from the ~1.5 Hz rate that we observed in the auditory modality. We believe indeed that our results represent an important contribution to the question of the existence of auditory oscillators, specific to periodic temporal attention in our case.

"Through six interrelated behavioural experiments, we reveal the existence of a limited sampling capacity of temporal attention, which is moreover sensory-specific." Given the structure of the tasks, i.e. the manipulation of distractor density in order to achieve certain levels of performance, one could argue that the paper is actually examining the ability of the brain to entrain an oscillator to a periodicity that might then guide attention, rather than an ongoing oscillator that is sampling its sensory environment, irrespective of what the sensory input is doing. It isn't clear to me how the experiments distinguish these two possibilities, and, as noted below, there is reason to believe that, at least for the auditory stimuli, the arrhythmic properties of the sequences created by the distractors are hampering the entrainment of a periodic attention process, especially if the periodicity is farther away from the preferred frequency. It is unclear whether the "sensory" input to the model also contained the distractors. It seems that the best way to model the behavior would be to feed into the model, on a trial-by-trial basis, the stimuli that participants actually heard.

Concerning the two alternative interpretations proposed by the reviewer, we don't claim that our paradigms distinguish between these two possibilities. Only neuroimaging evidence could, in our opinion, highlight the functional networks responsible for this beat-selective preference. That is why we always mention temporal attention in a sensory-specific way, and state in the discussion:

p. 16: "These specific rhythmic sampling rates would thus emerge from the specific configuration of large-scale neural networks encompassing sensory (in addition to attentional and motor) regions."

Next, we agree with the interpretation of the reviewer that the presence of distractors hampers "the entrainment of a periodic attention process, especially if the periodicity is farther away from the preferred frequency". As also discussed below, we added a sentence to better highlight the role of the distractors in our paradigm:

p. 5 and 18: "This interleaved delivery of sensory events forced participants to track the beat throughout the entire duration of the sequence while minimizing the interference of aperiodic events."

Finally, concerning the model, we clarified what we did in the Results section:

p. 12: "The external stimulus (S) was a purely periodic rhythm (i.e. without distractors), which varied in frequency to mirror our different experimental conditions (between 0.3 and 3.8 Hz)."

We chose to entrain the model with a purely periodic external oscillator (and not with the actual stimuli). We believe that it is more generic way to highlight a frequency selectivity. Moreover, the model needs much more iterations than human participants to show any entrainment behaviour. In our case the simulation lasted 1e4 seconds (vs. <12 seconds for human participants). We would thus need a more complex model to deal separately with the initialisation part provided in the experiment by the 3 first (periodic) reference tones, and the subsequent part where targets and distractors are co-occurring.

Methodological concerns:

Several methodological details about the construction of the trials are lacking:

First, it is unclear what the variability was in the number of possible events in the sequence. It is written that, "trials had pseudo-random durations (~2 to 10s) but included at least four targets and lasted at least 2 seconds." Given the different tempi, different trial durations would be expected for fixed numbers of events as a result, but the implication of the authors' statement is that the number of events varied also. What was the range of the number of possible events, and was this range the same for all tempi? Were the three reference tones taken into account when calculating the range of reported trial durations?

In our study, because we tested different tempi, the number of target events and the duration of the sequences were different across conditions. To avoid possible biases, we fixed a minimal duration and a minimal number of target events (that does not include the three reference tones). In the auditory modality, a trial was built with at least 4, and at most 22 target tones (excluding the first third reference stimuli and the deviant) and had a duration between ~2 and 12s. In the visual modality, because the slowest beat frequency is much slower than in auditory task, a trial was built with at least 4, at most 22 targets (without the first third reference stimuli and the deviant) and had a duration between ~2 and 20s. We thank the reviewer for pointing out this imprecision, and added this piece of information in the Method section:

p. 19: “Trials had pseudo-random durations (~2-12 s) but included in each condition at least four targets (and up to 22; reference tones and deviant excluded) and lasted at least 2 seconds.”

And for the visual modality:

p. 20: “Due to the presence of slower tempi, trials had pseudo-random durations of ~2-20 s.”

Second, on pg. 18, the magnitude(s) of the temporal deviation of the last event about which an on/off-the-beat judgment had to be made is not specified, but is absolutely critical to know. Was it a fixed amount or a proportion of the beat-frequency? If it was a proportion, then depending on tempo there could be an issue of temporal discrimination limits at higher rates (having nothing to do with attention). Across trials, could off-beat events either precede or follow the expected onset time, or were they consistently either early or late? Figure 1a implies that there was a range of off-beat times both before and after the expected event time, but this needs to be described clearly in the text.

The magnitude of the temporal deviation of the last event was indeed proportional to the beat-frequency. Off-beat deviants could occur within +/- half of the beat frequency of the trial. They could thus occur early or late relative to the beat. To be clearer we added the following sentence in the manuscript:

p.19: “When the deviant was off beat, it appeared randomly within a window corresponding to one beat period and centred around the expected beat.”

We chose this strategy as otherwise it would have been impossible to compare different beat frequencies with a single paradigm, as the investigated range is wide (>*5 between the slowest and fastest tempi). However, we are confident that our results do not reflect an absolute threshold in the temporal discrimination capacity of participants, as otherwise we would have observed a linear relationship between beat frequency and performance accuracy (with best performance for lower tempi). However, to confirm that the fluctuations of performance observed across tempi could not be explained by the position of the deviant, we conducted a GLMM analysis and confirmed that the beat frequency was significantly contributing to the model, even when taking into account this the magnitude of the temporal deviation of deviant target. We added two paragraphs, in the Method and Result sections, to describe this additional analysis:

Methods section:

*p. 22: “**Generalized linear mixed model.** A generalized linear mixed model (GLMM) regression analysis was performed in R (glmer function) on the passive auditory experiment (exp. 1) to characterize the extent to which performance was impacted by the nature of the deviant (on- or off-beat), its distance relative to the beat (in ms, in relative value) and the beat frequency. To do so, performance was analysed with these three variables used as predictors and participant number used as random factor.”*

Results section:

p. 6: “We also investigated whether this result could be explained by the position of the deviant relative to the beat. We conducted a GLMM analysis, in which the nature of the deviant (on- or off-beat) and its distance relative to the beat (in ms, in relative value) were entered as predictors, in addition to the beat frequency. These three variables were not correlated (all $R^2 < .13$), and each of them significantly contributed to the model (nature of deviant: $\text{Chi} = 18.2$; distance: $\text{Chi} = 13.2$; beat frequency: $\text{Chi} = -11.2$; all $p < .001$). This indicates that while the position of the deviant impacted performance accuracy, it could not explain the fluctuations of performance observed across tempi.”

Third, what proportion of trials were "on-beat" trials? Related to this is the question of why the authors are reporting proportion correct rather than the signal detection theory measure, d' , as would be more typical.

There is the same amount of on- and off-beat trials in our experiments. Due to this design, reporting proportion correct or d' is debatable (Kroll et al., 2002; Creelman and Macmillan, 1979; Micheyl, 2008; Yeshurun, 2008). In line with previous studies, which used an analogue design, we choose to report proportion correct (Rohenkohl et al., 2012; Cravo et al., 2013; Morillon et al., 2014; Morillon and Baillet, 2017; etc).

Kroll, N. E. A., Yonelinas, A. P., Dobbins, I. G. & Frederick, C. M. Separating sensitivity from response bias: Implications of comparisons of yes-no and forced-choice tests for models and measures of recognition memory. *J. Exp. Psychol. Gen.* **131**, 241–254 (2002).

Creelman, C. D. & Macmillan, N. A. Auditory phase and frequency discrimination: A comparison of nine procedures. *J. Exp. Psychol. Hum. Percept. Perform.* **5**, 146–156 (1979).

Micheyl, C., Kaernbach, C. & Demany, L. An Evaluation of Psychophysical Models of Auditory Change Perception. *Psychol. Rev.* **115**, 1069–1083 (2008).

Yeshurun, Y., Carrasco, M. & Maloney, L. T. Bias and sensitivity in two-interval forced choice procedures: Tests of the difference model. *Vision Res.* **48**, 1837–1851 (2008).

Cravo, A. M., Rohenkohl, G., Wyart, V. & Nobre, A. C. Temporal Expectation Enhances Contrast Sensitivity by Phase Entrainment of Low-Frequency Oscillations in Visual Cortex. *J. Neurosci.* **33**, 4002–4010 (2013).

Morillon, B., Schroeder, C. E. & Wyart, V. Motor contributions to the temporal precision of auditory attention. *Nat. Commun.* **5**, 1–9 (2014).

Morillon, B. & Baillet, S. Motor origin of temporal predictions in auditory attention. *Proc. Natl. Acad. Sci.* **114**, E8913–E8921 (2017).

Fourth, did distractors occur between the last target event and the to-be-judged event?

The answer is definitively yes. We added a comment in the article to clarify this point:

p.18: “This design implies that the last inter-stimulus interval (ISI) did not always correspond to the beat period of the trial. In particular, distractors could occur between the last target tone and the deviant.”

Fifth, Figure 3b indicates that the majority of participants had a "difficulty" level set to fewer than 1 distractor per beat, which means that there were many targets between which no

distractor would occur (particularly for vision). What therefore determined whether a distractor would be placed between any given pair of targets? Figure 3a shows a highly atypical sequence, in that there is at least one distractor between each pair of targets. However, only one participant attained a performance level of 1 distractor/beat in the visual condition. Please show a more representative trial.

The reviewer is right that when the difficulty level is below 1, there are beat periods for which no distractors are presented. We updated figure 3A, to provide a more realistic example of the visual paradigm.

Sixth, I think the answer is no, but please state explicitly whether deviants were constrained to fall on a metric grid? Given that the "target" and "distractor" tones had the same frequency (660 Hz?), non-metric distractors would be expected to have a *profound* impact on the ability to entrain to the tactus. What this issue highlights is the fact that this paper is more about the ability of aperiodic events to disrupt an endogenous oscillation than it is about the ability of an endogenous oscillation to facilitate perception. Previous studies of temporal attention have manipulated the timing of the target about which a judgment is made in order to determine how temporally constrained the focus of attention is, e.g. Barnes & Jones (2000); Jones et al. (2002). The fact that a different aspect of periodic temporal attention (its resilience to distractors) is being examined in this paper isn't really made clear.

Deviants (the last tone/Gabor of the trial) were not constrained to fall on a metric grid. We clarified this in the Methods section.

p. 19: "When the deviant was off beat, it appeared randomly within a window corresponding to one beat period and centred around the expected beat."

Concerning the main part of the comment, we fully comply with the reviewer's interpretation that our paradigm forces participants to ignore aperiodic events embedded in a periodic pattern. To make that clear in the manuscript we added the following sentence, p. 5 and 18:

"This interleaved delivery of sensory events forced participants to track the beat throughout the entire duration of the sequence while minimizing the interference of aperiodic events."

Relatedly, almost 50 years of auditory scene analysis research suggests that the frequency separation of the "target" and "distractor" tones should have a profound effect

on the ability to maintain attention at the beat frequency, and therefore on the results. If the metronomic "targets" were to segregate into their own stream, and participants can attend to that stream, would a similar profile across beat frequencies be observed? I'm not sure it would. This isn't to say that there wouldn't still be a preferred beat frequency (tempo), but if all of a sudden performance were to increase at non-optimal rates, e.g. 2.9 Hz, what would that have to say about periodic attention at a single preferred rate?

The reviewer highlights an important point of auditory physiology, namely the combination of temporal and spectral cues in auditory perception. As the auditory cortex is tonotopically organized, spectral (or feature-based) attention is more efficient than temporal attention to selectively focus (e.g., Wollman and Morillon, 2018). Accordingly, if target and distractor streams had a different spectral profile in our paradigm, the task would be much easier. However, our hypothesis is that in a spectrotemporal task (vs. temporal in the present manuscript), the optimal rate would still be around 1.4 Hz in the auditory modality. Lakatos and colleagues have shown how temporal and spectral cues combine into a spectrotemporal filter mechanism to optimise perception. In particular, they showed that the mechanism was similar in spectrotemporal tasks (Lakatos et al., 2013) and in purely temporal tasks (O'Connell et al., 2015). In the later article, they showed that when no spectral cues were present, temporal modulations were still occurring in spectrally specific regions of A1, in the form of a spectrotemporal modulation filter. It suggests that temporal modulations are always spectrally selective, and that the present paradigm already drives a spectrotemporal filter mechanism. We now added this set of ideas in the Discussion section, as follow:

p. 15: "Temporal attention is known to operate along the topographical dimension of the sensory system of interest^{75,76} (spectral in audition, spatial in vision), and to synergistically interact with information present on such dimension to optimize perception^{77,78}. Here, as sensory streams were providing clear and invariant spectral (in audition) or spatial (in vision) information, the impact of temporal attention on performance accuracy is thought to be optimal."

Cited references:

- Wollman, I. & Morillon, B. Organizational principles of multidimensional predictions in human auditory attention. *Sci. Rep.* **8**, 1–11 (2018).
- O'Connell, M. N. *et al.* Multi-scale entrainment of coupled neuronal oscillations in primary auditory cortex. *Front. Hum. Neurosci.* **9**, 1–16 (2015).
- Lakatos, P. *et al.* The Spectrotemporal Filter Mechanism of Auditory Selective Attention. *Neuron* **77**, 750–761 (2013).

pg. 19. "Following this short training session, participants performed a psychophysical staircase where the density of distractors was the varying parameter. The staircase was set to obtain 75 % of categorization performance." Was the distractor density value obtained for 2 Hz applied to all of the other beat frequency conditions? If it was set based on 2 Hz, should the curves in Figures 1c and 3c not be passing through 0.75? It appears that it is pretty close in 1c, but not in 3c.

Indeed, the distractor density value obtained at 2 Hz was applied to all the other beat frequency conditions. And the 2 Hz condition was not tested in the subsequent (main)

experiment (1.7 and 2.2 Hz were tested). However, the reviewer is right that in the visual modality, the performance level around 2 Hz is a bit lower than in the auditory experiment. This is due to the approximative nature of the staircase (only 10 minutes to estimate a perceptual threshold), and to the fact that throughout the experiment, participants progress. To compensate for this latter point, we tried to adjust the difficulty level by taking into account the expected learning rate during the main experiment. This could explain the difference that the reviewer pointed out between the auditory and visual modalities. Indeed, it might be that in the auditory modality participants progressed more during the experiment than in the visual modality.

There is another point. Because participants were much better in the auditory task (Fig. 3B), in other terms because the density of distractors in auditory sequences was globally higher than in visual sequences, it was easier to adjust this density precisely than in the visual modality.

In any case, this is not a central point in our study. Indeed, it is the global pattern across tempi that requires our interest. And difficulty level was mainly adjusted to avoid chance or ceiling values across tempi.

Minor points:

pg. 3 "A succession of events appearing too fast, or two stimuli too close in time, are situations typically difficult to attend to." This seems to be conflating the ability to discriminate individual events, attending to individual events, and attending to a stream of events. For example, it is easy to attend to a roll performed on a snare drum, even though one may not be focusing attention on the sound made by each hit on the drum.

This example was indeed confusing. We thus modified the sentence to:

p.3 "A stream of events appearing too fast or in a temporally disorganized way are situations typically difficult to attend to."

pg. 5 "The comparison of conditions revealed significant fluctuations in performance (% correct responses) across beats (repeated-measures ANOVA: $F(7,196) = 17.6$, $p < .001$; Fig. 1c)." Presumably by "across beats" you mean "across tempi" or "across beat frequencies" as opposed to individual beats within a trial.

Definitively yes. We changed the term "beat" by "tempo" in line to a previous comment of the reviewer.

p.5: "The comparison of conditions revealed significant fluctuations in performance (% correct responses) across tempi [...]."

pg. 5 "They moreover had an inverse U-shape profile" Unclear what "they" is referring to. Individual subject performance profiles across tempi?

We modified this sentence by the following one:

p.5: "The profile of performance across tempi had moreover an inverse U-shape profile, which could be properly approximated with a third-degree polynomial function [...]."

pg. 6 "These results suggest that during auditory perception temporal attention presents an optimal sampling rate, around 1.2 Hz." Very similar to the sentence two sentences before this one. Not sure what is gained by restating it.

We thank the reviewer for pointing out this redundancy in the text. We simply removed this second occurrence from the article.

pg. 6 "we thus orthogonalized beats and stimulus duration" I am assuming that by "beats" you again mean tempi. Because the meaning of the word beat is ambiguous, I encourage the authors to use the more conventional term, tempo, or beat frequency to refer to the different rates at which individual beat-keeping events are presented.

As stated above, we changed in many instances the term "beat" by "tempo" or "beat-frequency" throughout the manuscript, in line to a previous comment of the reviewer.

pg. 13 Reference to Fig. 6b-c is probably supposed to be Fig. 5b-c.

We thank the reviewer for pointing out this typographical error. We have double-checked our reference to the figures throughout the manuscript.

pg. 16 "Accordingly, participants were overall much more accurate in auditory than visual temporal attention (Fig. 3b)" It's unclear how Figure 3b shows this. Figures 1c and 3c would suggest that, if anything, participants were more accurate in the visual conditions.

We meant by this sentence that the difficulty (indexed by the proportion of distractors in a trial) is much higher in the auditory than the visual task. In other words, for an identical difficulty level, a participant would be worst in the visual task. Fig 3c represents the difficulty set for participants in the two modalities. To clarify this point, we added next to the sentence cited above the following one:

p.15: "Specifically, less distractors per beat had to be inserted in the visual sequences to obtain similar levels of performance."

References:

R. Barnes and M. R. Jones (2000). "Expectancy, attention, and time." *Cognitive Psychology* 41(3): 254-311.

C. Drake, M. R. Jones and C. Baruch (2000). "The development of rhythmic attending in auditory sequences: attunement, referent period, focal attending." *Cognition* 77(3): 251-288.

B. K. Hurley, L. K. Fink and P. Janata (2018). "Mapping the dynamic allocation of temporal attention in musical patterns." *Journal of Experimental Psychology: Human Perception and Performance* 44(11): 1694-1711. <http://doi.org/10.1037/xhp0000563>

M. R. Jones, H. Moynihan, N. MacKenzie and J. Puente (2002). "Temporal aspects of stimulus-driven attending in dynamic arrays." *Psychological Science* 13(4): 313-9.

E. W. Large, J. A. Herrera and M. J. Velasco (2015). "Neural Networks for Beat Perception in Musical Rhythm." *Frontiers in Systems Neuroscience* 9(159). <http://doi.org/10.3389/fnsys.2015.00159>

E. W. Large and M. R. Jones (1999). "The dynamics of attending: How people track time-varying events." *Psychological Review* 106(1): 119-159.

Reviewer #2 (Remarks to the Author):

This paper reports a series of six behavioral psychophysical experiments on auditory and visual temporal attention using a beat discrimination task. The results suggest that optimal sampling rates are higher in audition than in vision and that a secondary motor tracking task facilitates auditory temporal discrimination but impairs visual discrimination. The authors present a simple neural network model that can fit the data well.

Evaluation

Comparing the psychophysics of auditory vs. visual perceptual processing is useful, and the finding of opposite effects of a secondary motor tracking (i.e., tapping) task on vision vs. audition is possibly the most interesting result of this study. At this stage I am still somewhat ambivalent in my evaluation of the contribution of this paper. In my view, the task-specificity of modality-specific effects would need to be highlighted more, and there are some aspects of the design that limit a strict comparability of performance across the senses.

Comments

According to the notion of modality appropriateness, audition is specialized for temporal processing, whereas vision is specialized for spatial processing. Hence, when examining temporal discrimination processes, audition should be superior to vision, whereas when examining spatial discrimination it should be the reverse data pattern. This has been shown, repeatedly, in studies on crossmodal visual-auditory spatial attention and also crossmodal temporal attention. In my view, this issue of task-specificity of modality-specific effects need to be discussed in more detail.

We thank the reviewer to open an interesting discussion with regard to our results. We agree that the temporal dimension is often more associated with the auditory than the visual modality. However, in both modalities, temporal information is secondary compared to the coding dimension of the sense organ, namely tonotopy for audition and retinotopy for vision. Accordingly, recent works show that temporal information must be coupled to spectral (in audition) or spatial (in vision) information to optimize perception (Wollman and Morillon 2018; Rohenkohl et al., 2014). We placed our investigation in this line of research – by providing clear spectral (always the same tones) and spatial (always the same location) information – to investigate whether temporal attention had an optimal sampling rate, and whether it was associated to an amodal or a modality-specific underpinning. We now clarified our viewpoint on this issue in the Discussion section as follow:

p. 15: “Temporal attention is known to operate along the topographical dimension of the sensory system of interest (spectral in audition, spatial in vision), and to synergistically interact with information present on such dimension to optimize perception. Here, as sensory streams were providing clear and invariant spectral (in audition) or spatial (in vision) information, the impact of temporal attention on performance accuracy is thought to be optimal.”

We moreover cite articles by other groups that specifically investigated the sampling capacities of visual spatial (or feature-based) attention.

p. 15: “Recent studies have revealed the rhythmic nature of sustained attention, showing that spatial^{44,56-62} or featured-based⁴⁴ attention samples visual stimuli rhythmically, tethered by the phase of a theta (4-8 Hz) neural oscillation.”

References cited in the response:

Wollman, I. & Morillon, B. Organizational principles of multidimensional predictions in human auditory attention. *Sci. Rep.* **8**, 1–11 (2018).

Rohenkohl, G., Gould, I. C., Pessoa, J. & Nobre, A. C. Combining spatial and temporal expectations to improve visual perception. *J. Vis.* **14**, 1–13 (2014).

Moreover, the finding of sensory visual impairment when coupled with motor tapping would be more impressive if the reverse “dissociation” could be demonstrated as well, that is, impaired auditory spatial discrimination performance whilst visual spatial attention is improved. Currently, we have a single finding, and it is hard to know whether this tells us about sensory systems, task characteristics, or the interaction of both (I suspect the latter). Hence, while I believe that the present data are very interesting, the potential contribution could be further strengthened by discussion of (and possibly data on) the issue of whether the task demands spatial vs. temporal discrimination. Currently, the authors do not offer a strong theoretical account as to why this specific interaction is observed. (The fact that a neural network model can fit the data well shows us that the data are reasonably systematic but does not help us too much to “explain” the data pattern from a cognitive point of view. Such a discussion with respect to how sensory codes and motor codes interact in a domain- and task-specific way would also help to evaluate the novelty value of the present set of results.

We also comply with the reviewer supposition that our results reflect the interaction between temporal attention and sensory systems. We discussed that in the discussion section, e.g.:

p. 15/16: “However, independently of the overall accuracy effect, our results highlight a sensory-specific constrained sampling of temporal regularities, rather than an amodal optimal beat frequency at which temporal attention operates. These specific rhythmic sampling rates would thus emerge from the specific configuration of large-scale neural networks encompassing sensory (in addition to attentional and motor) regions^{85,86}.”

Regarding the motor involvement, it is unclear to us how motor involvement could be investigated in a purely spatial task, viz. without any temporal dimension. In line with the active sensing framework, we see motor activity as temporal in essence, and thus principally associated with temporal attention (Schroeder et al., 2010; Morillon et al., 2014; Morillon et al., 2019). We now more explicitly refer to this framework in the discussion section:

p. 16: “Second, we reveal that the quality of motor tracking directly benefits performance accuracy in auditory attention, but negatively impacts it in visual attention (Fig. 2f and 4e). This was related to the fact that the temporal alignment between motor and attention fluctuations leading to optimal performance differed between modalities (Fig. 2f and 4e). In

an active sensing framework, motor dynamics can act in concert with attention to temporally structure the activity of sensory cortices and shape perception^{13,33,41,93}. Our results thus suggest that the delay for optimal coordination between motor and attentional fluctuations differs across modalities, with a better synchronization in the auditory modality.”

Finally, it is true that our model does not offer a cognitive explanation of our results. If we refer to the three levels of Marr (Vision, 1982), our model provides an explanation at the implementation level, but not at the functional level.

Cited references:

Schroeder, C. E., Wilson, D. A., Radman, T., Scharfman, H. & Lakatos, P. Dynamics of Active Sensing and perceptual selection. *Curr. Opin. Neurobiol.* **20**, 172–176 (2010).

Morillon, B., Schroeder, C. E. & Wyart, V. Motor contributions to the temporal precision of auditory attention. *Nat. Commun.* **5**, 1–9 (2014).

Morillon, B., Arnal, L. H., Schroeder, C. E. & Keitel, A. Prominence of delta oscillatory rhythms in the motor cortex and their relevance for auditory and speech perception. *Neurosci. Biobehav. Rev.* (2019).

My impression is that the comparability of performance across the senses needs even more discussion. One aspect to be discussed is the individual titration of presentation rates, and possibly another would be the (possibility of) demonstration of psychophysical equivalence in terms of perceived stimulus intensity, etc. I am not meaning this “overcritically”, but I am just mentioning that it is really difficult to interpret “simple interaction” effects across modality. Perhaps this could be mentioned somewhere.

As mentioned above, in our study we maximized the sensory-specific information (spectral in audition, spatial in vision), to be able to properly investigate the temporal dimension. Thus, our stimuli are well above perceptual threshold, and it would not be relevant to estimate stimulus intensity.

Second, we comply with the reviewer’s concern that the individual titration (quantified by the number of distractors per beat period) was not well discussed in the previous version of our manuscript. Indeed, it highlights that participants have overall better temporal attention capacities in the auditory than the visual modality (which is well in line with previous cross-modal studies). We now state in the Discussion section:

p. 15: “Our paradigm used transient stimuli which are known to be more suited to auditory than visual perception^{76–78}. *Indeed, the visual modality is ecologically more precise for capturing movement whereas audition is more adapted to transient stimuli*⁷⁹. *Accordingly, participants were overall much more accurate in auditory than visual temporal attention (Fig. 3b)*^{77,80–84}. *Specifically, less distractors per beat had to be inserted in the visual sequences to obtain similar levels of performance.*”

In any case, this is not a central point in our study. Indeed, it is the global pattern across tempi that requires our interest. And difficulty level was mainly adjusted to avoid chance or ceiling values across tempi.

The authors argue that flexibly focusing in time and its limits has “never been investigated”, but this is probably overstated because there are studies on global vs. local focus of auditory attention (e.g., work by List or by Sanders, etc.) and even on switching the attentional focus between global and local auditory patterns. This could be mentioned (and the statement could be softened or specified accordingly).

We think that this argument does not concern our study. It seems that the reviewer refers to articles focussing on the effect of priming in global vs. local auditory attention studies (Justus and List, 2004; Sanders and Poeppel, 2007; Sanders and Astheimer, 2008), which are not in relation with a specific attentional dynamic (e.g. at 1.5 Hz). Indeed, these studies do not bring to light preferred temporal dynamics underlying the processing of relevant information over time. Rather, they investigate the hierarchical chunking of sensory information for fixed (short or long) temporal periods. We thus believe that the studies mentioned by the reviewer and our present work are complementary but non overlapping.

Cited references:

Justus, T. & List, A. Auditory attention to frequency and time: An analogy to visual local-global stimuli. *Cognition* 98, 31–51 (2004).

Sanders, L. D. & Poeppel, D. Local and global auditory processing: Behavioral and ERP evidence. *Neuropsychologia* 45, 1172–1186 (2007).

Sanders, L. D. & Astheimer, L. B. Temporally selective attention modulates early perceptual processing: Event-related potential evidence. *Percept. Psychophys.* 70, 732–742 (2008).

p. 6, line 139: Being no methods expert, I am still wondering whether the dfe for paired welsh t test can be 38 if it refers to a study with $n = 20$? At least this makes me wonder what experiment and which data is being analyzed here and which are the two conditions to be compared. Perhaps a little rewriting could help the reader (or at least this particular reader) to be able to follow better the description of the results.

We thank the reviewer for pointing out this error. Indeed, the adequate value was 19. We doubled-checked the degrees of freedom (and all the statistical values) throughout the manuscript.

p. 7, line 165: a BF of .32 and .33 closely corresponds to $1/3$, so is this taken as evidence for the H_0 ? A verbal paraphrase would help here in order to know what the authors would like to conclude from such statistics.

To clarify this point, we now state:

p.8: “Again, Bayes factor values provide significant evidence for the “null” hypothesis (no difference of optimal tempi).”

p. 8, line 169: The headline in bold makes a statement that seems to be contradicted by the last sentence of the same paragraph. Perhaps this could be clarified.

Indeed, the term “rhythmicity” was ambivalent. We changed it by the term that we use throughout the paragraph, namely “quality of motor tracking”:

p.8: “Overall, these results indicate that while the optimal rate of rhythmic movements and of auditory temporal attention is similar on average, there is no direct mechanistic relation between the quality of motor tracking and of auditory temporal attention.”

p. 9, line 217: Detail: did not revealed -> reveal

Done. We thank the reviewer for pointing out this typographical error.

p. 10, line 238: t(14) -> please describe the conditions/participants included in this comparison.

Indeed, there was again an error of typography. The dfe is 19 relative to the number of participants.

p. 11, line 276; theses analyses -> these

Done.

Reviewer #3 (Remarks to the Author):

Review of “Attention to rhythms: sensory-specific constrained sampling of temporal regularities” by Zalta, Petkoski, & Morillon

In six experiments the authors investigate the rhythmic behavior of temporal attention in three different modalities (visual, auditory, and motor). The paradigm they employed consisted of a sequence of stimuli presented at different frequencies. After the first three stimuli, additional distractor stimuli were presented at various temporal proximities. The participants' task was to determine whether the very last stimulus was either on or off-beat in comparison with the first three reference stimuli. By systematically varying the frequency of the stimuli they were able to determine that the most optimal beat frequency was about 1.4 Hz for auditory stimuli, 0.7 Hz for visual, and 1.7 Hz for motor stimuli, reflecting the modality-specific sampling rates of sensory perception and temporal attention. In addition, they revealed that finger tapping along with the presented rhythm actually helped with auditory performance when it was between 1.3 and 2.2 Hz, thus close to the optimal/preferred frequency of both audition and motor control. In contrast, tapping along with visual stimuli had no effect on performance except at 1.66 and 2.2 Hz where it actually interfered with visual performance. Finally, the authors present a computational model consisting of three delay-coupled phase oscillators to explain the pattern of findings.

General evaluation: I really enjoyed reading this well-written paper, which presents highly interesting results regarding the rhythmic behavior of temporal attention within and across different modalities, confirming the modality-specific nature of attentional restrictions. The experiments are elegantly designed, and the analyses and interpretation of the results are well done.

We thank the reviewer for this enthusiastic evaluation of our work. This is much appreciated.

I only have a couple of questions/suggestions as detailed below:

Experiment 1 and 2 provided somewhat different estimates of the optimal frequency for audition (1.2 hz in exp 1 and 1.5 hz in exp 2). I might have missed it, but a discussion why this might be the case seems to be missing. A potential explanation might lie in individual/group differences, and for that reason it is a pity that the authors did not control or at least measure the amount of musical experience that their participants had. It is not hard to imagine that a trained musician must show a lot less variability in tapping behavior as well as how it interacts with temporal attention in the visual modality (which might interfere even more) and the auditory modality (which might be more beneficial).

This is an interesting point highlighted by the reviewer. The variability observed in our results can have multiple origins. One of them is that, contrary to the tapping behaviour, which can be estimated in <1 minute, the optimal sampling rate observed in the perceptual tasks is very difficult to highlight and necessitates >1 hour of experiment. We thus first believe that our perceptual measures are not as robust as the motor ones.

Second, we investigated whether the difference between experiments #1 and #2 (1.3 Hz vs. 1.5 Hz) was significant and found that it was not. We added this new analysis in the Results section:

p. 7: “Of note, the optimal beat frequency was also not significantly different across the two auditory experiments (exp. 1. vs. passive exp. 2: unpaired welsh t-tests: $t(48) = .66$, $p = .51$, Bayes factor = .34).”

We moreover did the same analysis for the visual experiments:

p. 10: “Of note, the optimal beat frequency was also not significantly different across the two visual experiments (exp. 3. vs. passive exp. 4: unpaired welsh t-test: $t(46) = -.07$, $p = .95$, Bayes factor = .29).”

Third, we actually had excluded professional musicians in our experiments. (We investigate the difference between musicians and non-musicians in other projects in the team, and hence routinely measure this competence in our preliminary questionnaires.) We clarified this in the Methods section:

p.18: “We did not select participants based on musical training and none of them were professional musicians.”

Line 203: Please rephrase/elaborate this sentence, as I initially got confused by the “The average difficulty level ...was significantly lower than in the auditory task”. That is, it took me some time to realize that the visual task was actually much harder than the auditory task and that the difficulty level thus needed to be set lower for the visual task (introducing fewer distractors) to obtain comparable levels of performance.

Indeed, this sentence could be confusing for the reader. We thus complemented it by the following one:

p.9: “In other words, less distractors per beat had to be inserted in the visual as compared to the auditory sequences to obtain similar levels of performance.”

In line 230 the authors argue that participants tended to tap too fast in all conditions of the visual task. One page later though (line 258) it is written that participants were not anticipating the beat but tapped in reaction to it. Please explain.

We thank the reviewer for pointing out this ambivalent explanation. We do not see these two sentences as contradictory. Indeed, in the visual modality participants tend to tap faster than the given tempo (Fig. S3a). On the other hand, they also tended to tap after the beat (Fig. S4b).

In order to clarify this issue, we added the following sentences in the manuscript:

Legend of Figure S2: “The precision is expressed as the ratio (in %) between the average tapping frequency and the tempo. In other words, it indicates whether the tempo of the tapping is faster (>100 %) or slower (<100 %) than the tempo of the sequence. The horizontal line indicates the ideal ratio.”

p.11: “In other words, participants tended to tap both faster (Fig. S3a) and later (Fig. S4b) than the beat.”

As the sequences are quite short (<20 second) and that participants tended to tap just slightly too fast (5%, see Fig. S3a), the two effects stay relatively independent.

In lines 276-280, it is argued that a synergistic modulation of sensory processing seems to rely on the temporal alignment between motor and attention fluctuations (i.e. 1.5 hz (aud) being closer than 1.7 hz (motor) than 0.7 (vision) to 1.7 hz), but that this does not explain why motor involvement positively impacts auditory but not visual attention. Do I understand correctly that the time-delay between stimulus and motor oscillator was indeed the crucial additional component in your model to explain the pattern of results?

Definitively the answer is yes. The last sentence of the Results section was in our opinion nicely detailing this point:

p.13: “In addition to the natural frequency of the sensory-specific temporal attention oscillator, which varied between modalities, and the coupling strength between motor and attention oscillators K_{M-A} , which varied between passive and tracking sessions, the time-delay between the stimulus (S) and the motor oscillator (M; τ_{S-M}) was crucial for reproducing the difference of results across modalities.”

To be clearer we added the following sentence in the Discussion section:

p. 16: “Second, we reveal that the quality of motor tracking directly benefits performance accuracy in auditory attention, but negatively impacts it in visual attention (Fig. 2f and 4e). This was related to the fact that the temporal alignment between motor and attention fluctuations leading to optimal performance differed between modalities (Fig. 2f and 4e). In an active sensing framework, motor dynamics can act in concert with attention to temporally structure the activity of sensory cortices and shape perception^{13,33,41,93}. Our results thus suggest that the delay for optimal coordination between motor and attentional fluctuations differs across modalities, with a better synchronization in the auditory modality.”

It is a pity that the authors did not look at the interactions between the visual and auditory modalities as well by having some conditions within rather than only between participants. It would be interesting to see whether an individual who performs well in the auditory task would perform either better or worse in the visual task. Would it be possible to make any predictions using the existing model?

It is an interesting issue that we unfortunately cannot directly investigate with our current dataset, as each experiment had its own pool of participants. Thus, we cannot analyse the interactions between the visual and auditory modalities at a within-subject level. Moreover, the model explains global dynamics of periodic temporal attention and does not focus on sensory-specific thresholds, nor on the nature of the implemented noise. So, the current model cannot make any predictions regarding the relation between modalities.

I miss information as to whether participants were able to see their own (tapping) finger during the tasks. Is it possible that the movement of the finger could have had a distracting influence on performance (more so in the visual than in the auditory task)?

Participants were asked to fixate the centre of the screen in both auditory and visual experiments. In the auditory experiments, they had to fix a cross in the middle of the screen. This is stated in the method section:

p. 18: “On each trial, participants had to fixate a cross, located at the centre of the screen, to get a constant visual stimulation.”

I’m surprised that no further instruction/restriction was needed on how slow participants were allowed to tap in the ‘as-slow-as-possible-free-tapping-task’.

We updated the manuscript to clarify this point, that was indeed underspecified. Indeed, the instruction was to tap ‘as rhythmically and as fast (or slow) as possible’. No other instruction was provided.

p.21: “In two additional conditions, participants were instructed to tap rhythmically as fast and as slow as possible, for 30 and 60 seconds, respectively.”

Figure 1 (and 3): It would be helpful to explicitly/visually indicate the reference stimuli, the mixture of target and distractor stimuli, and the final target stimulus.

We think that is as easy to recognise the final target stimulus by its different tone frequency (785 Hz) and colour (blue) for auditory and visual experiments respectively. Target and distractor stimuli are also easily discernible by their position relative to the beat (vertical grey lines). To clarify the figures without altering their readability, we now indicate the three reference stimuli at the beginning of the sequence, as follow:

Line 717: It is rather hard to recognize the dashed line in panel C, i.e., it is not clear that it is dashed.

We modified the legend and now refer to the vertical line.

Figure 2b: Figure seem to be cut off at the right edge.

We double checked that all the values are represented, especially in the high frequency range.

Line 758: How/where was threshold performance defined?

Indeed, the term “threshold” was ambiguous. We modified the legend of figure 3(b) with the following sentence:

*p. 33: “**B.** Comparison of the individual difficulty levels obtained during the staircase procedures (with a 2 Hz tempo) of the auditory (exp. 1) and visual (exp. 4) experiments.”*

Signed, Sander Martens

Reviewers' Comments:

Reviewer #1:

Remarks to the Author:

The clarifications added by the authors are helpful, and most of my remaining comments are minor. However, there are still a couple of substantial issues that are related to each other and seem to be at the core of what makes this paper a newsworthy contribution to an already substantial literature on temporal attention.

Perhaps the most important feature of this paper is the parsimonious coupled-oscillator model of the behavior. While the model results shown in Figure 5 are compelling, it would be very helpful to have a bit more description (including graphs) of the model dynamics. In response to my previous question about why the results could not be simulated on a per-trial basis, the authors responded that the comparison wasn't really possible because the model settled over a simulated 10,000 seconds. Such a long period to instantiate or affect the oscillator dynamics that give rise to behavior would be useless in the real world. I think the authors need to speak to this point a bit more and make a more convincing case that the model is capable of explaining real-time behavior, i.e. responding to real-time inputs including the distractors (which would serve to perturb the phase of the oscillator especially if the distractors are occupying the same sensory feature channel, e.g. frequency). All of the oscillator models to date have a settling time of at least a few seconds, but not 10,000 seconds!

Relatedly, I still think that the distinction between periodic temporal attention (conceived of as an ongoing intrinsic oscillatory process with a tuning curve and optimal "sampling" frequency) and the capacity for temporal attending not only at a single frequency, but also within a hierarchy of metric levels (integer multiples and subdivisions of a reference frequency), as models of the entrainment of layers of oscillators tuned along a continuum of frequencies allow, needs to be set up more clearly in the introduction and/or taken up in the discussion. The listing of a number of references to previous models doesn't actually engage meaningfully with what seem to be markedly different conceptions of how attention operates or is constrained to operate in time.

Minor comments:

line 18: it seems that "constrained" would be an even better word than "bounded"

line 85: "Three reference stimuli defining the isochronous tempo (or beat frequency) of the sequence" -> Three reference stimuli defining the tempo (or beat frequency) of the isochronous event sequence

line 88: "While on-beat stimuli were providing the beat" -> While on-beat stimuli were reinforcing the beat

line 209/210: "In other words, less distractors per beat had to be inserted ..." -> In other words, fewer distractor had to be inserted per beat ...

line 387: "less distractors" -> fewer distractors

The font size of the labels along the arrows in Figure 5A should be larger.

line 426/27 - The added sentence makes it sound like the participant selection process was agnostic to musical training, which was not the case. Participants with musical training were excluded. This should be stated clearly, along with the criterion for how much musical training was too much musical training.

Reviewer #2:

Remarks to the Author:

Generally, I believe that the authors have been responsive to the reviewers' comments and revised the manuscript satisfactorily, so that I am happy to recommend publication (perhaps after a final, minor revision, see below).

For the "tracking" experiments it would be good to specify whether the order of "passive" and "tracking" sessions was counterbalanced across participants or whether it was constant (if so, which was the starting condition)?

Motor tapping vs. motor tracking. Perhaps it would be helpful to specify this further in order to avoid any confusion (such as adding "free" vs. "forced"/"guided")

When interpreting correlations of measures it would be good to consider the individual reliabilities of the measures that were correlated. The reliability of the individual measures define the upper boundaries of any correlation that could be found. For example, how reliable is the estimate of CV (e.g., would there be a strong paired-samples correlation of the CVs if they are calculated for first vs. second half of the experiment or for odd vs. even trials). Such a consideration could help to further qualify the interpretation of such correlations.

Iring Koch (signed review)

Reviewer #3:

Remarks to the Author:

I'm happy with the changes that the authors made to the revised manuscript in response to the reviewers' comments including my own. I have no more suggestions except for three minor textual corrections:

P. 15, line 387: Specifically, less distractors.. -> Specifically, fewer distractors..

Please fix the lacking/incorrect author name abbreviation for ref 64.

The manuscript titles of ref 91 and 99 are in capitals only. Please correct this.

Signed, Sander Martens

Response to reviewers

We would like to thank the reviewers and the editor for their constructive and positive evaluation of our revised manuscript. In the following, we reply to the reviewers' comments in a point-by-point manner: Plain font: unabridged reviewer comments; Blue font: our reply. Changes in the manuscript are also highlighted in blue.

Reviewers' comments:

Reviewer #1 (Remarks to the Author):

The clarifications added by the authors are helpful, and most of my remaining comments are minor. However, there are still a couple of substantial issues that are related to each other and seem to be at the core of what makes this paper a newsworthy contribution to an already substantial literature on temporal attention.

We thank the reviewer for this positive feedback on our revised manuscript.

Perhaps the most important feature of this paper is the parsimonious coupled-oscillator model of the behavior. While the model results shown in Figure 5 are compelling, it would be very helpful to have a bit more description (including graphs) of the model dynamics. In response to my previous question about why the results could not be simulated on a per-trial basis, the authors responded that the comparison wasn't really possible because the model settled over a simulated 10,000 seconds. Such a long period to instantiate or affect the oscillator dynamics that give rise to behavior would be useless in the real world. I think the authors need to speak to this point a bit more and make a more convincing case that the model is capable of explaining real-time behavior, i.e. responding to real-time inputs including the distractors (which would serve to perturb the phase of the oscillator especially if the distractors are occupying the same sensory feature channel, e.g. frequency). All of the oscillator models to date have a settling time of at least a few seconds, but not 10,000 seconds!

We remind the reviewer that our manuscript is first and foremost an experimental article, composed of six interrelated behavioral studies. We however agree that the coupled-oscillator model nicely synthesizes these results and clarifies the key constraints governing the temporal alignment between motor and attention fluctuations. To better describe the dynamics of the model, we now have added a new panel on Figure 5:

We added the following legend to describe the new panel B:

p 29-30: “**B.** Example of dynamics of the auditory temporal attention oscillator (A; green) during presentation of an external beat (S; black) at 1.7 Hz or 3.8 Hz.”.

We also now justify why we used a simulation duration of 1e4 seconds, in the Method section. 1e4 seconds approximately corresponds to the entire duration of one experiment, summed across trials and participants. Indeed, it is important to notice that we fitted the model to the group- and trial-averaged behavioural data, i.e. data averaged over (at least) 20 participants performing the task for (at least) 2 hours each. Individual trials have no explanatory power in our paradigm. Indeed, we never analyse real-time behaviour, but compare across conditions (beat frequencies) the capacity of participants to perform a binary discrimination task at the end of each trial. The model being stationary (after maybe some very short transition, which is probably insignificant and can be discarded) and ergodic, averaging over trials is the same as averaging over time. What matters is the total number of events (periods) that was used to compute the statistics, and this number was (approximately) matched across behavioural and modelling data.

What the reviewer suggests is in essence to capture how the distribution of the distractors in the stimulus’ stream impacts trial-to-trial performance accuracy. To this end, a different class of model might be more appropriate, but this is out of the scope of the current study.

Relatedly, I still think that the distinction between periodic temporal attention (conceived of as an ongoing intrinsic oscillatory process with a tuning curve and optimal "sampling" frequency) and the capacity for temporal attending not only at a single frequency, but also within a hierarchy of metric levels (integer multiples and subdivisions of a reference frequency), as models of the entrainment of layers of oscillators tuned along a continuum of frequencies allow, needs to be set up more clearly in the introduction and/or taken up in the discussion. The listing of a number of references to previous models doesn't actually engage meaningfully with what seem to be markedly different conceptions of how attention operates or is constrained to operate in time.

We agree with the reviewer's concern and now clearly set up this distinction in the Discussion section:

p 17: "Previous models of beat perception and temporal attention processes mainly focus on the capacity of a dynamical system composed of coupled oscillators tuned along a continuum of frequencies to temporally attend not only at a single frequency, but also within a hierarchy of metric levels^{23,24,45-47}. Here, we instead developed a model in which periodic temporal attention is conceived as one ongoing intrinsic oscillatory process with an optimal sampling rate."

Minor comments:

line 18: it seems that "constrained" would be an even better word than "bounded"

Thank you for this good suggestion. We now state:

p 2: "Whether this function is constrained by an underlying rhythmic neural mechanism is unknown".

line 85: "Three reference stimuli defining the isochronous tempo (or beat frequency) of the sequence" -> Three reference stimuli defining the tempo (or beat frequency) of the isochronous event sequence

Done.

p 5: "Three reference stimuli defining the tempo (or beat frequency) of the isochronous event sequence preceded a mixture of on- and off-beat stimuli."

line 88: "While on-beat stimuli were providing the beat" -> While on-beat stimuli were reinforcing the beat

Done.

p 5: "While on-beat stimuli were reinforcing the beat, crucially, off-beat stimuli had a distracting influence."

line 209/210: "In other words, less distractors per beat had to be inserted ..." -> In other words, fewer distractor had to be inserted per beat ...

Done.

p 9: "In other words, fewer distractor had to be inserted per beat in the visual as compared to the auditory sequences to obtain similar levels of performance"

line 387: "less distractors" -> fewer distractors

Done.

The font size of the labels along the arrows in Figure 5A should be larger.

Indeed, the font size of the labels along the arrows in Figure 5A were quite little. We enlarged them to improve the readability. Please see above the updated figure.

line 426/27 - The added sentence makes it sound like the participant selection process was agnostic to musical training, which was not the case. Participants with musical training were excluded. This should be stated clearly, along with the criterion for how much musical training was too much musical training.

Actually, the participant selection process was agnostic to musical training. We only asked participants to fill a survey about their musical practice at the end of the experiment. We observed that none of the participants were professional musicians, this is probably due to the channel by which we recruited participants. To clarify this, we now state:

p 19: "We did not select participants based on musical training and a short survey made at the end of the experiment informed us that none of them were professional musicians."

Reviewer #2 (Remarks to the Author):

Generally, I believe that the authors have been responsive to the reviewers' comments and revised the manuscript satisfactorily, so that I am happy to recommend publication (perhaps after a final, minor revision, see below).

We thank the reviewer for this positive appreciation of our revised manuscript.

For the "tracking" experiments it would be good to specify whether the order of "passive" and "tracking" sessions was counterbalanced across participants or whether it was constant (if so, which was the starting condition)?

The order of "passive" and "tracking" sessions was indeed counterbalanced across participants. We clarified this in the Methods section, by adding the following sentence:

p 21: "In experiment 2, they also performed 1 tracking session (320 trials). The order of the sessions, passive and tracking, was counterbalanced across participants. Conditions (tempi) were pseudo-randomly alternating in blocs of 20 trials each."

Motor tapping vs. motor tracking. Perhaps it would be helpful to specify this further in order to avoid any confusion (such as adding "free" vs. "forced"/"guided")

Thank you for this suggestion. We now use 'free tapping' and 'guided tapping' throughout the manuscript to avoid any confusion.

When interpreting correlations of measures it would be good to consider the individual reliabilities of the measures that were correlated. The reliability of the individual measures define the upper boundaries of any correlation that could be found. For example, how reliable is the estimate of CV (e.g., would there be a strong paired-samples correlation of the CVs if they are calculated for first vs. second half of the experiment or for odd vs. even trials). Such a consideration could help to further qualify the interpretation of such correlations.

In the manuscript, we present interrelated experiments that provide complementary findings but also serve the purpose of replicating each other. For example, the observation of an optimal sampling rate of auditory temporal attention, observed around 1.4 Hz, is replicated in three experiments (exp. 1, 2 and 6).

To convince the reviewer of the reliability of our results, we also divided the data acquired in experiment 2 into *odd* (left panels) and *even* (right panels) trials. Results are reported in the figure below. The reviewer can appreciate that the results are very similar across these two clusters of trials, which further proves that the individual data are reliable.

Iring Koch (signed review)

Reviewer #3 (Remarks to the Author):

I'm happy with the changes that the authors made to the revised manuscript in response to the reviewers' comments including my own. I have no more suggestions except for three minor textual corrections:

We thank again the reviewer for his positive evaluation of our work.

P. 15, line 387: Specifically, less distractors.. -> Specifically, fewer distractors..

Please fix the lacking/incorrect author name abbreviation for ref 64.

The manuscript titles of ref 91 and 99 are in capitals only. Please correct this.

Done.

We thank the reviewer for pointing out these typographical errors.

Signed, Sander Martens

Reviewers' Comments:

Reviewer #1:

Remarks to the Author:

I have no further comments.

Reviewer #2:

Remarks to the Author:

The authors responded to my comments, and I recommend publication basically as.

Perhaps just a brief comment for clarification: my comment on reliability was referring to within-subject correlation (i.e., whether the data of the odd-numbered trials correlate with the data of the even-numbered trials; which is a measure of split-half reliability). So, even if the means averaged across the group look very similar, it is still possible that the intraindividual correlation is very low. (As an experimentalist myself I was very surprised to see this pattern in some of my own data.)

I am not at all questioning the robustness of the findings at the group level (i.e., replicability) but it is rather the intraindividual consistency in which I was interested.

But I am not going to dwell on that issue and simply recommend publication.

Signed review, Iring Koch